# POWERDRESS interacts with HISTONE DEACETYLASE 9 to promote aging in *Arabidopsis*

Xiangsong Chen[1,2], Li Lu[1,2], Kevin S Mayer[1,2], Mark Scalf[3], Shuiming Qian[1,2], Aaron Lomax[1], Lloyd M Smith[3], Xuehua Zhong[1,2]*

[1]Laboratory of Genetics, University of Wisconsin-Madison, Madison, United States; [2]Wisconsin Institutes for Discovery, University of Wisconsin-Madison, Madison, United States; [3]Department of Chemistry, University of Wisconsin-Madison, Madison, United States

**Abstract** Leaf senescence is an essential part of the plant lifecycle during which nutrients are re-allocated to other tissues. The regulation of leaf senescence is a complex process. However, the underlying mechanism is poorly understood. Here, we uncovered a novel and the pivotal role of *Arabidopsis* HDA9 (a RPD3-like histone deacetylase) in promoting the onset of leaf senescence. We found that HDA9 acts in complex with a SANT domain-containing protein POWERDRESS (PWR) and transcription factor WRKY53. Our genome-wide profiling of HDA9 occupancy reveals that HDA9 directly binds to the promoters of key negative regulators of senescence and this association requires PWR. Furthermore, we found that PWR is important for HDA9 nuclear accumulation. This study reveals an uncharacterized epigenetic complex involved in leaf senescence and provides mechanistic insights into how a histone deacetylase along with a chromatin-binding protein contribute to a robust regulatory network to modulate the onset of plant aging.

*For correspondence: xuehua.zhong@wisc.edu

**Competing interests:** The authors declare that no competing interests exist.

## Introduction

Age-dependent organ and tissue dysfunction is detrimental to all organisms. Leaf senescence is an integral part of the plant lifecycle. Although efficient senescence is important to increase plant viability in the next generation, premature senescence often causes a decrease in the yield and quality of crops (*Avila-Ospina et al., 2014*; *Distelfeld et al., 2014*; *Guo and Gan, 2014*). Thus, the knowledge of mechanisms underlying leaf aging has profound implications in many biotechnological applications, including increasing plant productivity and preventing post-harvest loss during transportation and storage. Regulation of leaf senescence is a complex process controlled by developmental and environmental signaling pathways (*Lim et al., 2007*; *Woo et al., 2013*; *Schippers, 2015*). Many senescence-associated genes (*SAGs*) and transcription factors have been identified (*Gepstein et al., 2003*; *Buchanan-Wollaston et al., 2005*; *Breeze et al., 2011*; *Guo and Gan, 2012*). However, their in vivo function in senescence remains largely unknown. For the relatively well-studied senescence-associated transcription factors, current knowledge of their function is mostly derived from knockout mutants, transgenic overexpressing plants, or identification of downstream target genes. Little is known how these transcription factors are regulated and function mechanistically in the global control of leaf senescence.

Epigenetic modification is an important gene regulatory mechanism in eukaryotic organisms and plays critical roles in diverse biological processes, including genome stability and integrity, normal growth and development, diseases, and stress responses (*Kawashima and Berger, 2014*; *Matzke and Mosher, 2014*; *Pikaard and Mittelsten Scheid, 2014*). Histone (de)acetylation plays

**eLife digest** The leaves of many plants turn yellow in the fall as nutrients are recycled to prepare for the winter months. However, if leaves age and yellow too early, it can limit how much energy the plant can harvest from light. Thus, it is crucial for plants to know when they should start the leaf aging process. This is also important for plant biologists because premature leaf yellowing can reduce both the yield and quality of crop plants.

Certain aging-related genes tightly control when and how leaves age. Like in many other organisms, plant DNA is packaged around proteins called histones. As such, one of the ways that plants regulate the activity of their genes is by chemically modifying the DNA or histones to alter how tightly the DNA is packaged. For example, to switch particular genes off, enzymes known as histone deacetylases remove an acetyl group from their histones. However, it is not clear how these enzymes know which genes to modify and how this helps to make sure that leaf aging happens at the appropriate time.

Chen et al. studied a histone deacetylase called HDA9 in a flowering plant named *Arabidopsis*. The experiments show that the HDA9 enzyme plays an important role in ensuring the leaves turn yellow at the right time. Without HDA9, the leaf aging process is delayed. HDA9 also needs the help of another protein called PWR that instructs HDA9 to remove acetyl groups from the histones of specific aging-associated genes in order to switch these genes off.

The next challenge is to understand how HDA9 and PWR sense developmental and environmental signals to trigger the histone modifications. It will also be important to decipher how this enzyme works with other regulators to trigger leaf aging at the right time.

important roles in genome expression, organization, and function through the coordinated activities of histone acetyltransferases and deacetylases (*Haberland et al., 2009*; *Wang et al., 2014*; *Verdin and Ott, 2015*). While acetylation is often associated with active transcription, histone deacetylases (HDACs) are generally considered transcriptional repressors that remove acetylation and induce chromatin compaction (*Verdin and Ott, 2015*).

HDACs are highly conserved enzymes in eukaryotes. The flowering plant *Arabidopsis thaliana* has eighteen annotated histone deacetylases that are grouped into three families: twelve RPD3-like (REDUCED POTASSIUM DEPENDENCE 3), two SIR2 (SILENT INFORMATION REGULATOR 2), and four plant-specific HD2 (HISTONE DEACETYLASE 2) based on sequence similarity and cofactor dependence (*Pandey et al., 2002*). Genetic studies have revealed the critical function of HDACs in crosstalk between plant genomes and the environment in plant responses to diverse stresses at the cellular and organismal levels (*Krogan and Long 2009*; *Kim et al. 2012*; *Pikaard and Mittelsten Scheid, 2014*; *Wang et al., 2014*). Certain HDACs (e.g. HDA6 and HDA19) also function in genome integrity and gene silencing (*Murfett et al., 2001*; *Probst et al., 2004*; *May et al., 2005*; *Earley et al., 2006*; *Vaillant et al., 2007*; *Tanaka et al., 2008*; *Pontvianne et al., 2013*). Interplays between HDACs and other epigenetic modifications have also been documented. For example, HDA6 is important for DNA methylation (*Aufsatz et al., 2002*; *Earley et al., 2010*; *To et al., 2011*; *Liu et al., 2012*; *Blevins et al., 2014*; *Stroud et al., 2013*). Functional disruption of HDACs often causes pleiotropic abnormalities in plant growth and development (*Krogan and Long, 2009*; *Kim et al., 2012*; *Pikaard and Mittelsten Scheid, 2014*; *Wang et al., 2014*). For instance, early studies using an antisense approach to knockdown histone deacetylases suggest a potential role of histone deacetylation in leaf senescence (*Tian and Chen, 2001*). HDA6 is implicated in leaf senescence by downregulation of two *SAGs* in the loss-of-function mutants (*Wu et al., 2008*). However, the underlying mechanism through which HDA6 and other HDACs act in leaf senescence is unknown.

Histone deacetylase 9 (HDA9) is a RPD3 type deacetylase, closely related to mammalian HDAC3 (*Pandey et al., 2002*). Previous genetic mutational studies have established important roles of HDA9 in flowering (*Kim et al., 2013*; *Kang et al., 2015*), seed germination (*van Zanten et al., 2014*), and salt and drought stress (*Zheng et al., 2016*). However, the composition, regulation, and mechanistic action of HDA9 are unknown.

In this study, we report the identification and characterization of a previous uncharacterized repressive complex containing HDA9 and a SANT domain-containing protein POWERDRESS (PWR) as novel regulators of leaf senescence. HDA9 promotes the onset of age-related and dark-induced leaf senescence by regulating the expression of genes involved in senescence. Our genome-wide profiling of HDA9 occupancy reveals that HDA9 directly binds to the promoters of key negative regulators and this binding requires PWR. PWR physically interacts with HDA9 and loss-of-function *pwr* mutants phenocopy *hda9*, indicating that this complex is biologically relevant. Furthermore, we demonstrate that PWR is important for HDA9 nuclear accumulation as HDA9 protein level is significantly reduced in the nucleus in *pwr* mutants. Thus, we propose that PWR acts at multiple levels to regulate HDA9 function. To our knowledge, this is the first genome-wide study on targeting and regulating mechanism for a histone deacetylase in plants. Together, this study reveals an uncharacterized epigenetic complex involved in leaf senescence and provides mechanistic insights into how a histone deacetylase along with a chromatin-binding protein contribute to a robust regulatory network to promote the onset of plant aging.

## Results

### PWR interacts with HDA9

To gain mechanistic insights into HDA9 action, we identified the protein complex associated with HDA9 by performing immunoaffinity purification followed by multidimensional protein identification technology mass spectrometry (IP-MS). We generated *Arabidopsis* transgenic plants expressing *HDA9*-3xFLAG driven by the native *HDA9* promoter in *hda9* mutant background (pHDA9::HDA9-3xFLAG/*hda9*, abridged as HDA9-FLAG, *Figure 1—figure supplement 1A*). HDA9-FLAG rescued the dwarf phenotype of *hda9* (*Figure 1—figure supplement 1B*)(*Kang et al., 2015*), suggesting that HDA9-FLAG is functional in vivo. As a control, the same purification was performed in parallel with wild type Col-0 (WT). Our IP-MS analysis revealed 51 unique HDA9 peptides and also identified a peptide corresponding to a known HDA9-interacting protein AHL22 (*Yun et al., 2012*) (*Figure 1A*, *Figure 1—source data 1*). Besides HDA9, the most enriched protein in our MS is a SANT domain-containing protein POWERDRESS (PWR) with 27 unique peptides (*Figure 1A*). To confirm the PWR interaction, we generated transgenic plants expressing *PWR*-3xFLAG driven by its endogenous promoter (pPWR::PWR-3xFLAG, abridged as PWR-FLAG) in WT background. Reciprocal IP-MS of PWR-FLAG also purified HDA9 (*Figure 1A*, *Figure 1—source data 1*). To further validate the HDA9-PWR interaction, we performed co-immunoprecipitation (co-IP) experiments in F1 *Arabidopsis* plants expressing both HA-tagged HDA9 and FLAG-tagged PWR. When we pulled down PWR with anti-FLAG beads, we detected the co-precipitation of HDA9 with an anti-HA antibody (*Figure 1B*).

Our IP-MS also showed the co-purification of the WRKY53 transcription factor with HDA9 (*Figure 1A*). WRKY53 is induced at the early stage of leaf senescence and promotes the onset of senescence (*Miao and Zentgraf, 2007*). To confirm HDA9-WRKY53 interaction, we expressed and purified GST tagged full-length WRKY53 protein from *E. coli*, incubated with HDA9 protein purified from *Arabidopsis* HDA9-FLAG transgenic plants, and performed an in vitro GST pull down assay. HDA9-FLAG was pulled down by GST-WRKY53 but not GST itself (*Figure 1C*), suggesting that WRKY53 interacts with HDA9.

### Loss-of-function HDA9 and PWR mutations induce H3 hyperacetylation in vivo

The physical association of PWR with HDA9 led us to propose that PWR is important for HDA9 activity and function in vivo. Previous studies revealed that HDA9 is critical for deacetylation of H3K9 (H3K9ac) and H3K27 (H3K27ac) in vivo (*Kim et al., 2013*; *van Zanten et al., 2014*). Accordingly, our immunoblotting assays revealed increased H3K9ac and H3K27ac levels in *hda9* mutant, but not in the HDA9-FLAG complementation plants (*Figure 2A*, *Figure 2—figure supplement 1*). Given that PWR physically interacts with HDA9, we examined whether PWR is important for H3K9ac and H3K27ac deacetylation. Similar as *hda9*, loss-of-function *pwr* mutant induces global H3K9 and H3K27 hyperacetylation (*Figure 2B*). We next investigated the genetic interaction between HDA9 and PWR by generating an *hda9 pwr* double mutant and found substantial increases of H3K9ac and H3K27ac in *hda9 pwr* double mutants compared to WT (*Figure 2B*). Consistent with our data that

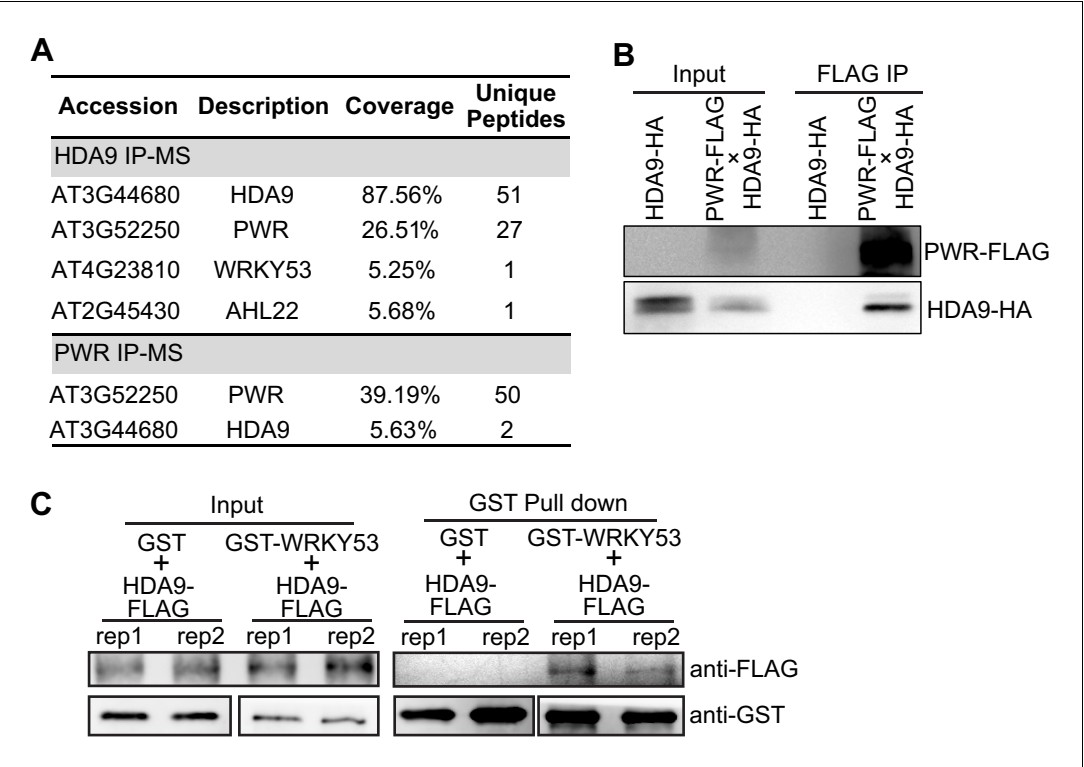

**Figure 1.** HDA9 interacts with PWR and WRKY53. (**A**) Summary of partial proteins associated with HDA9 and PWR identified by affinity purification and mass spectrometry analysis. Coverage indicates the percentage of full-length protein covered by identified unique peptides. Unique peptides indicate the number of identified peptides that are mapped to an individual protein. (**B**) Co-immunoprecipitation of HDA9 and PWR using *Arabidopsis* F1 plants expressing both HDA9-HA and PWR-FLAG. (**C**) In vitro pull down assay of GST-WRKY53 and HDA9-FLAG. Two technical replicates were performed (rep1 and rep2). GST protein serves as a control.

The following source data and figure supplement are available for figure 1:

**Source data 1.** List of proteins identified by IP-MS in HDA9 and PWR.

**Figure supplement 1.** HDA9-FLAG protein is functional in *Arabidopsis*.

PWR functions together with HDA9 (*Figure 1*), no significant differences in the increase of H3K9ac and H3K27ac levels were noted between single and *hda9 pwr* double mutants (*Figure 2B*).

To identify the specific hyperacetylated regions in *hda9* and *pwr*, we performed H3K27ac chromatin immunoprecipitation followed by sequencing (ChIP-seq) in *hda9* and *pwr* mutants. Consistent with the immunoblotting, we identified 11,372 and 7687 H3K27ac increased peaks in *hda9* and *pwr*, respectively (*Figure 2C*). We found that ~90% (6901 peaks) of *pwr* hyperacetylated peaks overlapped with those increased peaks in *hda9* (*Figure 2C*), suggesting that PWR and HDA9 target at similar genomic regions. Genomic distribution analysis of these hyperacetylated peaks showed that most of them are located in genic regions (97% for *hda9* and 96% for *pwr*, respectively) (*Figure 2D*), near the transcription start sites (*Figure 2E,F*). Together, these results suggest that PWR and HDA9 mediate deacetylation of H3K27ac at similar genomic regions.

## HDA9 preferentially binds promoters of actively transcribed genes

HDACs are generally considered transcriptional co-repressors associated with silent genes. To identify the in vivo binding pattern of HDA9, we determined the genomic occupancy of HDA9 using ChIP-seq in plants expressing HDA9-FLAG. The ChIP-seq was performed in parallel with WT. HDA9 is highly enriched in gene-rich euchromatic regions, but depleted in repeat-rich centromeric

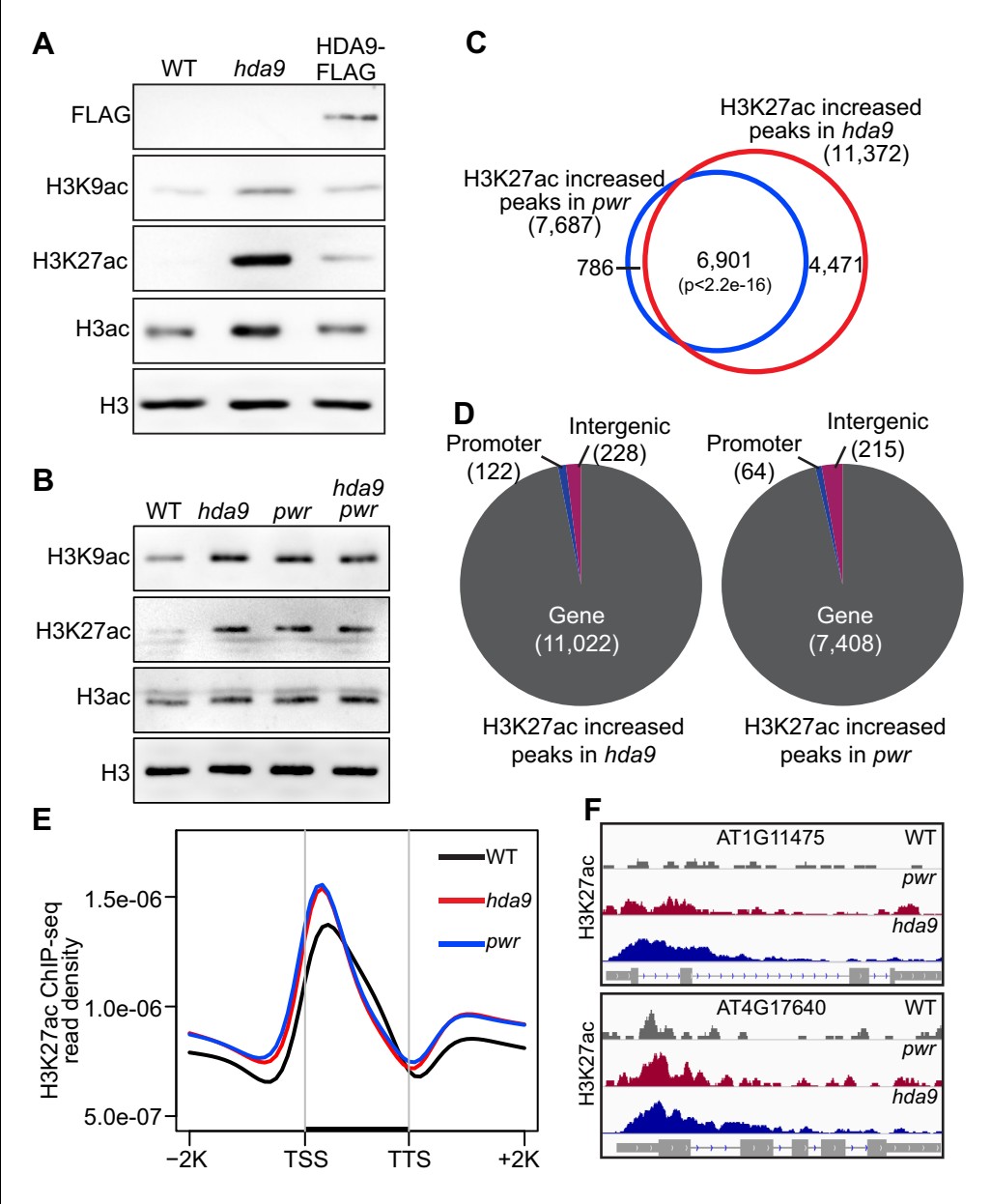

**Figure 2.** Loss-of-function *hda9* and *pwr* mutants induce H3K9 and H3K27 hyperacetylation. (**A**) Immunoblots of histone acetylation marks in *hda9*, *pwr,* and HDA9-FLAG early senescent leaves. (**B**) Immunoblots of histone acetylation marks in *hda9*, *pwr*, and *hda9 pwr* early senescent leaves. (**C**) Overlap of H3K27ac increased peaks in *hda9* and *pwr* identified by ChIP-seq. Fisher's exact test was used to calculate the p-value. (**D**) Genomic distribution of H3K27ac increased peaks in *hda9* and *pwr*. (**E**) Metaplots of the H3K27ac distribution on genes in WT, *hda9*, and *pwr*. Black bar in the X-axis represents all genes in the genome. TSS, Transcription Start Sites; TTS, Transcription Terminal Sites; −2K and +2K represent 2 kb upstream of TSS and 2 kb downstream of TTS, respectively. The Y-axis represents read density of H3K27ac ChIP-seq. (**F**) Browser snapshots of representative loci with increased H3K27ac in *hda9* and *pwr*.

The following figure supplement is available for figure 2:

**Figure supplement 1.** HDA9 preferentially removes acetylation on histone H3 tail in vivo.

heterochromatin (*Figure 3A*). Further analysis identified a total of 9489 binding peaks (p=1e-03) corresponding to 8232 genes (*Figure 3B*, *Figure 3—source data 1A*). The majority of HDA9 binding peaks (6515 or approximately 69%) were located in promoter regions (*Figure 3B*). Similar observations were reported in a genome-wide profiling of mammalian HDACs (*Wang et al., 2009*). We next examined the relationship between HDA9 binding and gene expression levels. We divided all 28,000 *Arabidopsis* genes equally into five groups based on their expression levels and correlated them with HDA9 binding. Surprisingly, we found that HDA9 is preferentially enriched in the promoters of active genes but not silent genes (*Figure 3C*). We also found that HDA9 bound genes showed significantly higher expression than the average expression levels of all genes (*Figure 3D*). To further examine the relationship between HDA9 and active genes, we correlated HDA9 binding with DNase I hypersensitive sites that are generally associated with accessible chromatin states (*Zhang et al., 2012*). We found co-localization between HDA9 binding and DNase I hypersensitive sites in gene promoters (*Figure 3E*, *Figure 3—figure supplement 1A*). Thus, HDA9 is associated with active genes.

To determine the biological significance of our defined HDA9 binding sites, we correlated the HDA9 binding genes with their H3K27ac levels in *hda9*. By comparing the H3K27ac levels over HDA9 and non-HDA9 bound genes, we found that HDA9 bound genes showed much higher increased H3K27ac levels in *hda9* compared to non-HDA9 bound genes (*Figure 3F*). Similarly, HDA9 binding is highly correlated with *pwr* induced H3K27 hyperacetylation (*Figure 3F*), suggesting that PWR and HDA9 mediate deacetylation of H3K27ac at similar genomic regions.

We next searched for putative DNA-binding motifs for HDA9 binding peaks using the DREME algorithm (*Bailey, 2011*) and identified seven significantly enriched consensus motifs (cut off p<1e-07 and minimum of 1200 peaks) (*Figure 3—figure supplement 1B*). Among them, 1233 HDA9 binding peaks (13%) showed the significant enrichment of G-box (CACGTG) motif (p=1.8e-129) (*Figure 3G*) that is present mostly in the promoter regions of genes and recognized by transcription factors (*Menkens et al., 1995*). PIF4/5 is one of the G-box binding proteins previously shown to be involved in leaf senescence (*Sakuraba et al., 2014*). In addition, we identified a W-box motif (TTGAC/T), recognized by WRKY family transcription factors (*Rushton et al., 2010*), as another putative HDA9 recognition motif (1328 peaks, p=1.9e-8) (*Figure 3G*), consistent with our observation that WRKY53 is co-purified with HDA9 (*Figure 1A,C*).

## PWR and HDA9 act in the same pathway to promote age-related and dark-induced leaf senescence

The HDA9-WRKY53 interaction and enrichment of WRKY binding motif in HDA9 binding sites led us to investigate a potential role of HDA9 in leaf senescence. We examined the leaf yellowing in *hda9* T-DNA knockout mutants and WT plants. After five weeks of growth, we observed the tips of older leaves in WT yellowed sooner than those of *hda9* (*Figure 4A*). By measuring the number of days from germination to leaf tip yellowing, we found that *hda9* mutant leaves became senescent at 39 days, significantly later than WT (35 days, p=4e-06) and the HDA9-FLAG complementation plants (37 days, p=0.025) (*Figure 4B*). Consistent with the role of HDA9 in the early stage of senescence, we found slightly elevated HDA9 protein in early senescent leaves (*Figure 4—figure supplement 1*).

Leaf senescence can also be induced by environmental stresses such as darkness (*Lim et al., 2007*). To examine the role of HDA9 in dark-induced leaf senescence, we analyzed leaf yellowing of the fifth and sixth rosette leaves detached from the plants of WT and *hda9*. Dark-induced leaf yellowing was attenuated in the *hda9* mutant relative to WT (*Figure 4C*). Consistent with the visible phenotype, *hda9* mutant leaves had greater total chlorophyll content than WT after dark treatment (*Figure 4D*). Thus, HDA9 promotes the onset of both age-related and dark-induced leaf senescence.

Given that PWR physically interacts with HDA9, we examined whether PWR also contributes to leaf senescence. Loss-of-function *pwr* mutants phenocopy *hda9*, showing delayed senescence of both naturally aged and dark-treated leaves (*Figure 4C–F*). Furthermore, we found that there was no noticeable difference in the degree of delayed leaf yellowness between the *hda9 pwr* double and the respective single mutants (*Figure 4C*). The similar leaf senescence phenotypes of *hda9*, *pwr*, and *hda9 pwr* mutants were further supported by their similar retention of chlorophyll content (*Figure 4D*). These observations indicate that HDA9 and PWR act in the same pathway to promote leaf senescence.

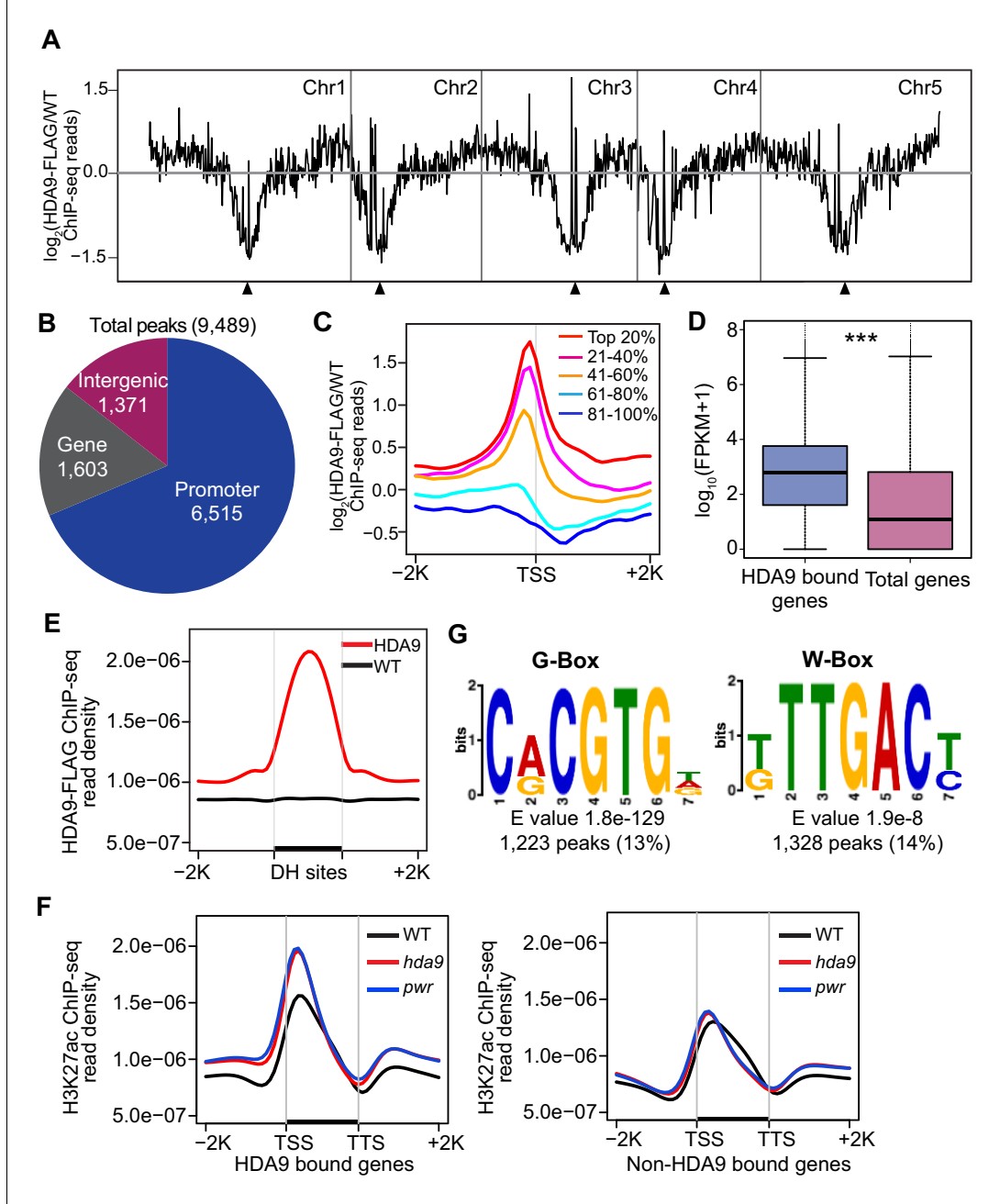

**Figure 3.** HDA9 binds to promoters of active genes. (A) Chromosomal views of HDA9 distribution on five chromosomes. The Y-axis represents the $\log_2$ value of HDA9-FLAG ChIP-seq reads relative to those of untagged WT control. Chr1, Chr2, Chr3, Chr4, and Chr5 represent chromosomes 1 to 5, respectively. Black triangles indicate the location of centromeric regions. (B) Genomic distribution of HDA9 binding peaks. (C) Metaplots of HDA9 binding levels on genes. Total genes were divided evenly into five groups based on their expression level in WT. Top 20% indicates the 20% genes with highest expression level, 81%–100% indicates the 20% genes with lowest expression level. The Y-axis represents the $\log_2$ value of HDA9-FLAG ChIP-seq reads relative to those of untagged WT control. −2K and +2K represent 2 kb upstream and downstream of TSS, respectively. (D) Box plots of the average expression levels of HDA9 bound genes and total genes. The Y-axis indicates $\log_{10}$ value of FPKM + 1. FPKM, Fragments Per Kilobase of transcript per million mapped reads. Bars within the boxes represent the mean values. \*\*\*p<0.001. (E) Metaplots of HDA9 binding on previously identified DH (DNase I Hypersensitive) sites in HDA9-FLAG and untagged WT control. Black bar in the X-axis represents DH sites. The Y-axis represents the read density of HDA9-FLAG ChIP-seq reads. (F) Metaplots of H3K27ac on HDA9 bound genes and non-HDA9 bound genes in WT, *hda9*, and *pwr*. Black bar in the X-axis represents genes. The Y-axis represents read density of H3K27ac ChIP-seq reads. (G) Representative DNA motifs identified in HDA9 binding sites by DREME.

*Figure 3 continued on next page*

*Figure 3 continued*

The following source data and figure supplement are available for figure 3:

**Source data 1.** HDA9 binds to active genes.
**Figure supplement 1.** HDA9 binds to open chromatin regions with known DNA motifs.

## PWR and HDA9 regulate the expression of the same group of genes involved in leaf senescence

The onset of leaf senescence is often accompanied by increased expression of senescence-associated genes (*SAGs*) and decreased expression of senescence downregulated genes (*SDGs*)

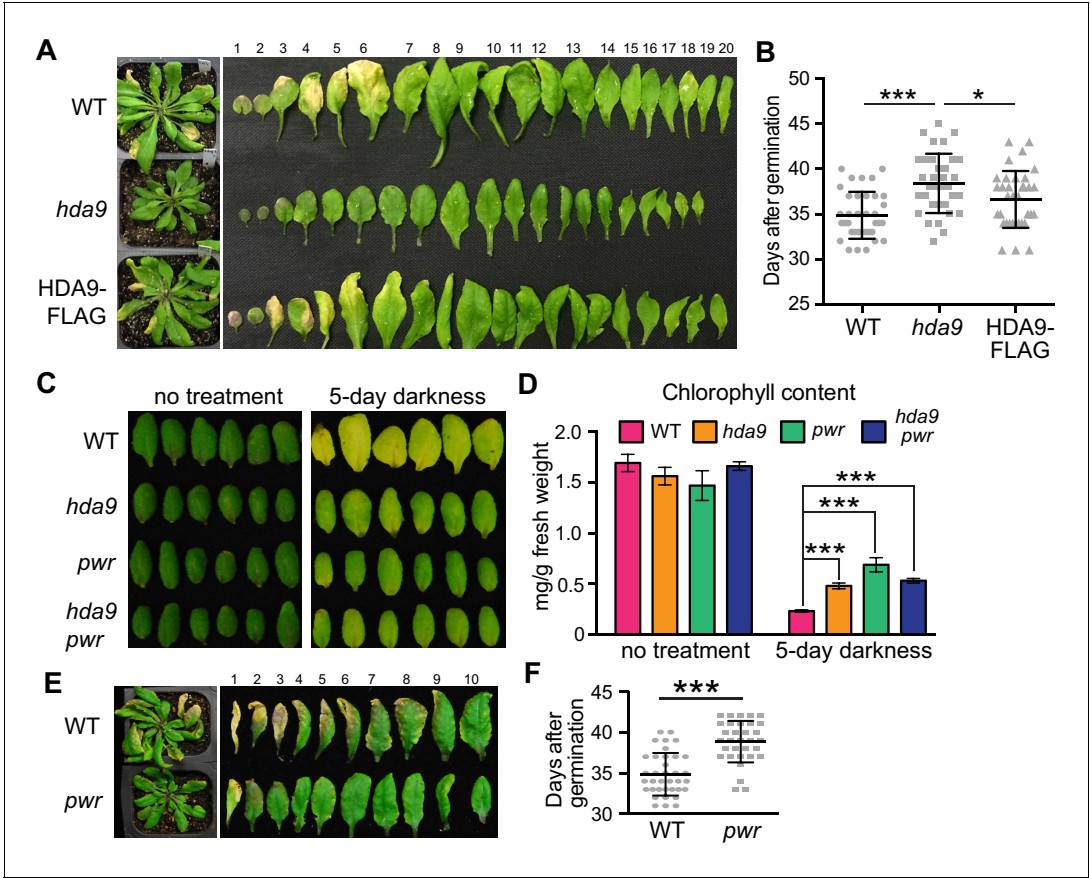

**Figure 4.** HDA9 and PWR act in the same pathway to promote leaf senescence. (A) Phenotypic analysis of leaves from 5-week-old plants of wild type (WT), *hda9*, and HDA9-FLAG plants expressing HDA9-FLAG driven by the native HDA9 promoter in *hda9* mutant background. Rosette leaves were numbered from bottom to top with the first leaf being the oldest and 20th being the youngest. (B) Quantification of days from germination to onset of leaf senescence in WT, *hda9* and HDA9-FLAG. Points (round, square, or triangle) represent the number of days for an individual plant to reach onset of senescence. Error bars represent standard deviation for at least 30 tested plants. (C) Dark treatment of the 5th and 6th leaves detached from 3-week-old plants of *hda9*, *pwr*, and *hda9 pwr* double mutants. (D) Chlorophyll content measurement of leaves from (C). Error bars represent a standard deviation for three biological replicates. (E) Leaf senescence phenotype of 5-week-old *pwr* mutant. The oldest ten leaves are shown. (F) Quantification of days from germination to the onset of leaf senescence in WT and *pwr*. Points (round or square) represent the number of days for an individual plant to reach onset of senescence. Error bars represent standard deviations for at least 30 tested plants. Student's t-tests were used to calculate the p values. *p<0.05, ***p<0.001.

The following figure supplement is available for figure 4:

**Figure supplement 1.** HDA9-FLAG shows elevated protein accumulation in early senescent leaves.

(*Gepstein et al., 2003*; *Breeze et al., 2011*; *Brusslan et al., 2015*). Consistent with the delayed leaf senescence phenotype in *hda9*, we found the downregulation of hallmark *SAGs* including *SENESCENCE4* (*SEN4*), *SAG12,* and *SAG113* in *hda9* (*Figure 5A*). To further examine the role of HDA9 in senescence, we performed whole transcriptome analysis by mRNA sequencing (RNA-seq) in *hda9* mutant and WT of two week-old young leaves (YL) as well as early senescence (ES) leaves. A previous study identified differential regulation of 3474 *SAGs* and 2849 *SDGs* during different stages of leaf senescence (*Breeze et al., 2011*). Although the expression of 30% of the *SAGs* (1023 out of 3474) increased in senescing leaves of both *hda9* and WT compared to the young leaves (*Figure 5B*), the fold change was significantly less in *hda9* compared to WT (*Figure 5C*). Similarly, decreased expression levels of 575 *SDGs* (20% of 2849) were impaired in *hda9* senescing leaves (*Figure 5—figure supplement 1A,B*), suggesting that HDA9 regulates the expression of senescence-related genes to promote leaf senescence. Besides *SAGs* and *SDGs*, many abscisic acid (ABA) response genes known to promote the onset of leaf senescence, were significantly downregulated (*Figure 5—figure supplement 1C*), further supporting a role for HDA9 in leaf senescence.

We identified 782 upregulated and 656 downregulated genes in *hda9* early senescent leaves compared to WT (*Figure 5—figure supplement 1D*, *Figure 5—source data 1*). These differentially expressed genes showed enrichment in senescence related pathways including jasmonic acid (JA) and ABA response (*Figure 5—figure supplement 1E*). To further explore the biological significance of HDA9 binding, we examined the correlation between HDA9 binding and altered gene expression in *hda9*. We found 222 genes (~28%) were shown to be bound by HDA9 and significantly upregulated in *hda9* (*Figure 5D*, *Figure 3—source data 1B*), suggesting that they are directly regulated by HDA9. Of the 222 genes, 151 (~68%) showed increased H3K27ac in *hda9*, indicating a positive correlation between gene upregulation and H3K27ac increase of HDA9 targets (p=1.2e-10) (*Figure 5—figure supplement 1F*). Interestingly, the majority of HDA9 bound genes did not show significant expression change in *hda9* (*Figure 5D,E*). Similar observations are reported in human (*Wang et al., 2009*) and maize (*Yang et al., 2016*). It is possible that HDA9 is not a primary regulator of transcription at the majority of its targets. Another possibility is that other HDACs or chromatin/transcriptional repressors function together with HDA9 to regulate gene expression, and thus loss of HDA9 itself is insufficient to release the transcriptional repression of its targets. Given the function of HDA9 in leaf senescence, we sought to investigate whether HDA9 directly regulate genes involved in this process. By searching the 222 genes with HDA9 binding and upregulation in *hda9,* we found 11 genes with potential or known functions in senescence (*Figure 5—figure supplement 1G*), including catalase that protects cells from oxidative damage (CAT1) (*Du et al., 2008*), autophagy proteins that delay senescence and programmed cell death (APG9, ATG2, ATG8E and ATG13) (*Hanaoka et al., 2002*; *Yoshimoto et al., 2004*; *Suttangkakul et al., 2011*; *Wang et al., 2011b*), proteins that negatively regulate ABA signaling pathway known to promote senescence (NPX1, PLL5, AFP2, AFP4) (*Schweighofer et al., 2004*; *Huang and Wu, 2007*; *Garcia et al., 2008*; *Kim et al., 2009b*), BIK1 that negatively regulates the salicylic acid (SA) signaling pathway (*Veronese et al., 2006*), and WRKY57 (a WRKY family transcription factor) acts as a negative regulator of JA to prevent leaf senescence (*Jiang et al., 2014*).

To further dissect the functional relationship of PWR with HDA9 in leaf senescence, we performed RNA-seq in *pwr* and identified 887 upregulated and 860 downregulated genes relative to WT in ES leaves (Student's t test, p<0.05) (*Figure 5—figure supplement 1H*, *Figure 5—source data 2*). Of the affected genes in *hda9,* 277 out of 782 upregulated genes (38%) and 354 out of 656 downregulated genes (54%) showed up- or downregulation in *pwr*, respectively (*Figure 5F*, *Figure 5—source data 3*). The number is much larger than expected by chance (Fisher's exact test, p<2.2e-16). GO analysis of the overlapping genes showed enrichment in developmental and environmental stress and ABA signaling pathways (*Figure 5—figure supplement 1I*), indicating that PWR and HDA9 regulate the expression of the same group genes in leaf senescence. *NPX1*, one of the HDA9 direct targets, also showed significant upregulation in *pwr* (*Figure 5—source data 3*). Besides *NPX1*, we also found upregulation of *APG9* and *WRKY57* in *pwr*. However, they are not defined to be significant based on our significance criteria. To further examine their expression, we performed RT-qPCR with additional biological replicates and confirmed that *APG9* and *WRKY57* were significantly upregulated in *pwr* (*Figure 5G*). All together, these results together with the physical interaction of HDA9 and PWR (*Figure 1*) support the notion that PWR and HDA9 act in the same pathway to promote leaf senescence.

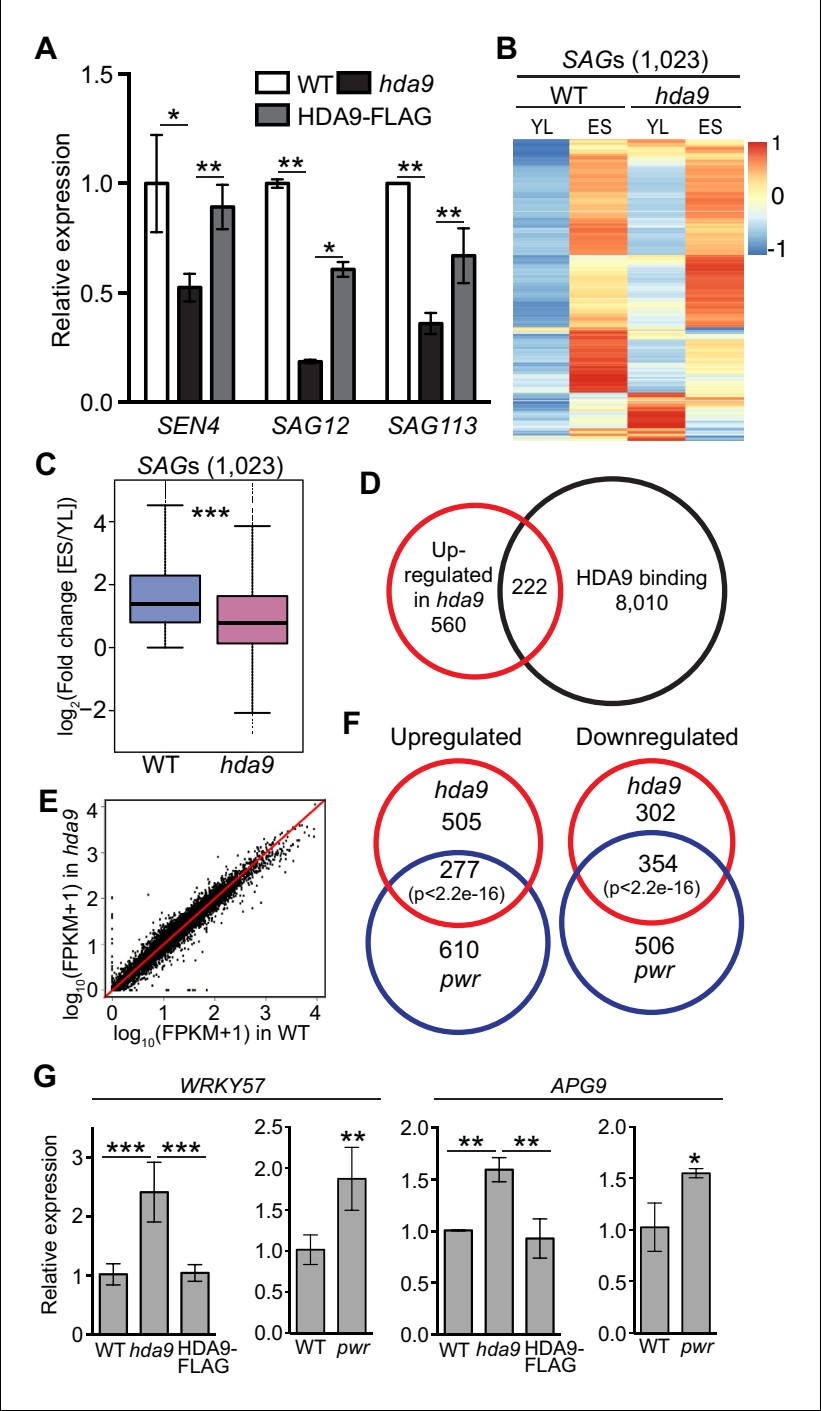

**Figure 5.** HDA9 and PWR regulate expression of the same group of genes involved in leaf senescence. (**A**) Expression of senescence marker genes in *hda9* by quantitative RT-PCR. Error bars represent a standard deviation from two biological replicates. (**B**) Heatmaps show expression of senescence-associated genes (*SAGs*) in young leaf (**YL**) and early senescence leaf (**ES**) in WT and *hda9*. The color bar on the right indicates the Z-score. (**C**) Boxplots show the less increased expression of *SAGs* in *hda9* than WT in ES. The Y-axis represents $\log_2$ value of fold change of expression levels of *SAGs* between ES and YL. (**D**) Overlap of upregulated genes in *hda9* and HDA9 bound genes. (**E**) Scatter plots show the expression of HDA9 bound genes in WT and *hda9*. (**F**) Overlap of differentially expressed genes in RNA-seq of *hda9* and *pwr*. Fisher's exact test was used to calculate the p-value. (**G**) Quantitative RT-PCR confirming the upregulation of *WRKY57* and *APG9* in *hda9* and *pwr*. Relative expression was calculated as relative to *ACTIN7*, and then normalized to WT. Error bars represent a standard deviation from two biological replicates. *$p<0.05$, **$p<0.01$, ***$p<0.001$.

*Figure 5 continued on next page*

*Figure 5 continued*

The following source data and figure supplement are available for figure 5:

**Source data 1.** Differentially expressed genes in *hda9*.
**Source data 2.** Differentially expressed genes in *pwr*.
**Source data 3.** List of overlapped genes showing differential expression in both *hda9* and *pwr*.
**Figure supplement 1.** HDA9 and PWR regulate similar group of genes involved in leaf senescence.

## HDA9 nuclear accumulation and chromatin association are dependent on PWR

We have confirmed PWR as a functional partner of HDA9 (*Figures 4* and *5*). To further dissect the molecular mechanism of PWR on HDA9 function, we examined whether PWR directly binds the same targets as HDA9. We performed ChIP-qPCR in PWR-FLAG plants and found that 9 of the 11 randomly chosen HDA9 bound loci showed significant enrichment of PWR (*Figure 6—figure supplement 1A*). Furthermore, PWR specifically binds to the same genomic regions within *APG9*, *WRKY57*, and *NPX1* where HDA9 are enriched (*Figure 6A*). Next, we asked whether PWR affected histone acetylation on the same targets of HDA9. We performed ChIP-qPCR and found that H3K27ac levels were significantly increased in *hda9* and *pwr* mutants at *WRKY57*, *APG9*, and *NPX1* (*Figure 6B*). To further examine whether *pwr* induced H3K27 hyperacetylation is correlated with HDA9 genome-wide binding, we compared the H3K27ac levels of HDA9 bound genes over non-HDA9 bound genes, and found that HDA9 bound genes showed a significantly higher increase of H3K27ac relative to non-HDA9 bound genes in *pwr* ($p<2.2e-16$) (*Figure 6C*). Together, these results suggest that PWR binds to the same genomic regions as HDA9 at HDA9 targets.

Given the loss-of-function *pwr* mutation induced global H3K9 and H3K27 hyperacetylation (*Figure 2*), one possible role for PWR is the recruitment of HDA9 to target loci. To test this hypothesis, we crossed an HDA9-FLAG line into the *pwr* mutant and confirmed that the overall HDA9 protein level was not affected in *pwr* (*Figure 6—figure supplement 1B*). We then performed ChIP-qPCR to determine the chromatin association of HDA9 in the absence of PWR. We found that the enrichment of HDA9 at *WRKY57*, *APG9*, *NPX1*, and four other randomly selected loci was substantially decreased in *pwr* (*Figure 6D*, *Figure 6—figure supplement 1C*), suggesting that HDA9 binding to these targets requires PWR. HDA9 needs to be imported in the nucleus for its histone deacetylation activity. The abolishment of HDA9 chromatin association in *pwr* promotes us to examine whether PWR is important for HDA9 nuclear accumulation. We performed a nuclear-cytoplasmic fractionation assay and found that HDA9 was present both in the cytoplasm and in the nucleus (*Figure 6E*). Interestingly, HDA9 accumulation in the nucleus was greatly reduced in *pwr* mutants compared to the plants with PWR (*Figure 6E*). The similar accumulation of HDA9 in the total extracts of WT and *pwr* indicates that PWR is important for HDA9 nuclear accumulation.

## Discussion

Epigenetic regulation plays important roles in many aspects of biological processes. Our study investigates the function and mechanism of epigenetic regulation in the aging process, an essential part of the plant lifecycle that has a big impact on agricultural productivity. This work uncovered a novel complex containing HDA9, PWR, and a transcription factor WRKY53 that act together to promote leaf senescence. We propose a model in which PWR facilitates the transport of HDA9 from the cytoplasm into the nucleus. In the nucleus, WRKY53 recruits PWR and HDA9 to W-box containing promoter regions. HDA9 catalyzes the removal of H3 acetylation marks and suppresses the expression of key negative regulators, which in turn induces the derepression of their downstream target genes to promote leaf senescence (*Figure 7*).

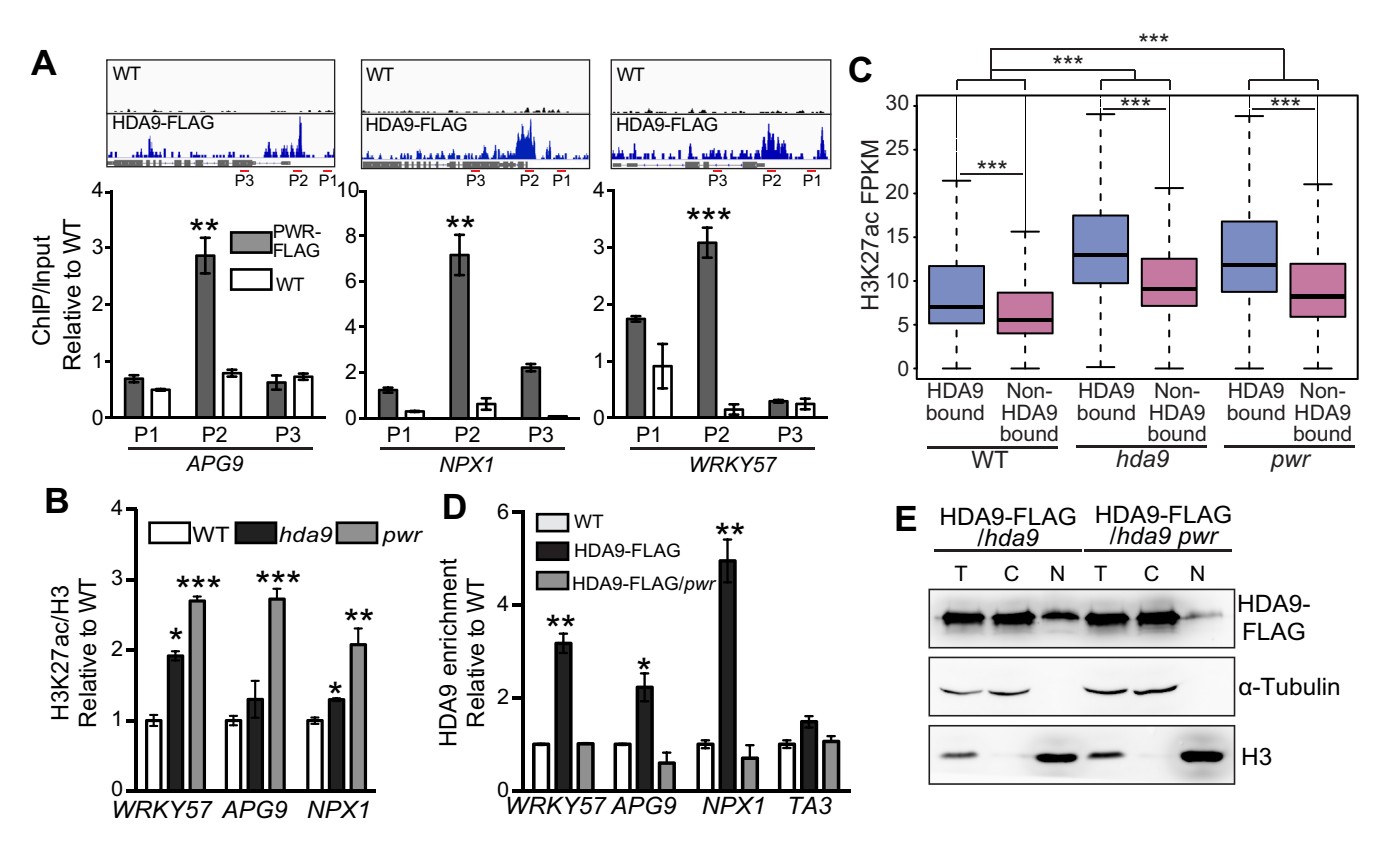

**Figure 6.** HDA9 nuclear accumulation and chromatin association are dependent on PWR. (A) ChIP-qPCR shows that PWR is enriched at the same genomic regions of HDA9 targets. Upper panel illustrates snapshots of HDA9 binding at *APG*, *NPX1*, and *WRKY57*. ChIP-qPCR value of PWR was normalized to WT control. Primer positions are indicated with P1, P2, and P3. Error bars represent a standard deviation from two biological replicates. (B) ChIP-qPCR shows H3K27ac levels at *WRKY57*, *APG9*, and *NPX1* in WT, *hda9*, and *pwr* mutants. ChIP-qPCR value of H3K27ac was normalized to WT control. Error bars represent a standard deviation from two biological replicates. (C) Box plots of H3K27ac levels on HDA9 bound genes and non-HDA9 bound genes in WT, *hda9*, and *pwr*. The Y-axis represents FPKM of H3K27ac ChIP-seq reads. Student's t test, ***p<0.001. (D) ChIP-qPCR shows HDA9 enrichment on *WRKY57*, *APG9*, and *NPX1* in HDA9-FLAG and HDA9-FLAG/*pwr* plants. *TA3* is a transposable element that serves as a negative control. Error bars represent a standard deviation from two biological replicates. *p<0.05, **p<0.01. (E) Detection of HDA9-FLAG protein in total (T), cytoplasmic (C), and nuclear (N) extracts in HDA9-FLAG/*hda9* and HDA9-FLAG/*hda9 pwr*.

The following figure supplement is available for figure 6:

**Figure supplement 1.** PWR is required for HDA9 recruitment to targets.

## HDA9 regulates multiple pathways to promote leaf senescence

Leaf senescence is a complex process regulated by multiple pathways. In this study, we found that many ABA-responsive genes were downregulated in *hda9* (*Figure 5—figure supplement 1C*), indicating that the ABA signaling pathway is impaired in *hda9* during leaf senescence. This is consistent with a previous study showing insensitivity to ABA-mediated seed dormancy and germination inhibition in *hda9* (*van Zanten et al., 2014*). ABA is known to promote leaf senescence (*Jibran et al., 2013*; *Khan et al., 2014*) and loss-of-function of the ABA receptor PYL8 or PYL9 causes delayed leaf senescence (*Lee et al., 2015*; *Zhao et al., 2016*). Our ChIP-seq and RNA-seq analyses reveal that neither PYL8 nor PYL9 is a primary target of HDA9. Instead, we found that HDA9 directly binds and represses two negative regulators of the ABA signaling pathway, *NPX1* and *AFP4/TMAC2* (*Figure 3—source data 1B*). NPX1 and AFP4/TMAC2 are proposed to act as negative regulators in ABA signaling because their overexpression reduces plant sensitivity to ABA (*Huang and Wu, 2007*; *Kim et al., 2009b*). Based on these findings, we propose that HDA9 promotes leaf senescence in

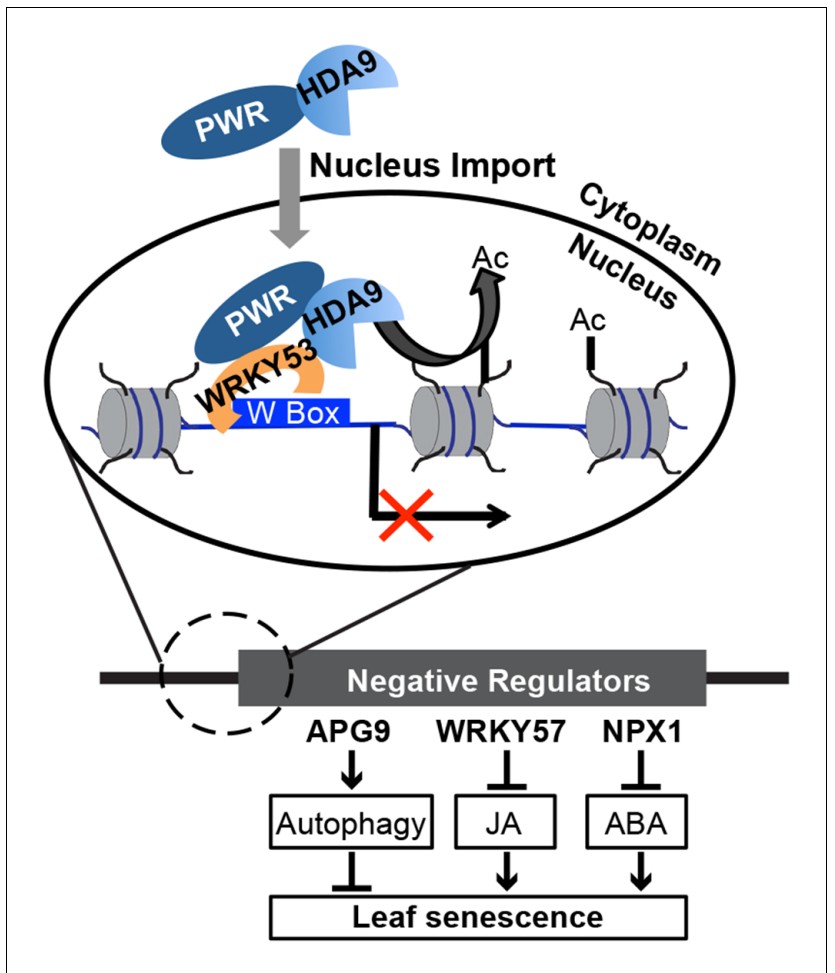

**Figure 7.** A working model for the biological functions and molecular mechanisms of HDA9 and PWR. In the cytoplasm, PWR forms complexes with HDA9 and is required for the transport of HDA9 from the cytoplasm into the nucleus. In the nucleus, PWR recruits HDA9 to W-box containing promoter regions likely with the help of WRKY53. HDA9 catalyzes the removal of H3 acetylation marks and suppresses the expression of negative senescence regulators (e.g. WRKY57, APG9, and NPX1), which in turn induces the derepression of their downstream target genes to promote leaf senescence.

part by repressing the negative regulators of the ABA signaling pathway (*Figure 7*). Besides ABA, one negative regulator of the JA pathway, WRKY57, is also shown to be a direct target of HDA9 (*Figures 5* and *6*, *Figure 3—source data 1B*). WRKY57 is a transcription factor that represses JA-induced leaf senescence (*Jiang et al., 2014*), suggesting that HDA9 may also regulate leaf senescence through the JA signaling pathway. Autophagy is an intracellular process for protein degradation and is associated with leaf longevity (*Avila-Ospina et al., 2014*). APG9 is an essential component of the plant autophagy pathway and *apg9* mutants display early leaf senescence (*Hanaoka et al., 2002*). Consistently, we found *APG9* and several other autophagy genes (*ATG8E*, *ATG2* and *ATG13*) are directly targeted and repressed by HDA9 (*Figures 5* and *6*, *Figure 3—source data 1B*).

Leaf senescence results in many cellular metabolic changes, including accumulation of oxidation products and reduction of antioxidant enzymes such as catalase (*Griffiths et al., 2014*). We found a significant enrichment of HDA9 at *CATALASE1 (CAT1)* and *CATALASE3* (*Figure 3—source data 1*), encoding enzymes that decompose hydrogen peroxide (*Du et al., 2008*). Our RNA-seq analysis also revealed a significant increase of *CAT1* expression in *hda9* (*Figure 5—source data 1B*), suggesting that HDA9 represses *CAT1* to allow the accumulation of oxidation during leaf senescence.

Interestingly, *CAT1* is reported to be a direct target of WRKY53 (*Miao et al., 2004*), consistent with the HDA9 and WRKY53 interaction (*Figure 1*). Thus, HDA9 appears to act in multiple pathways to promote leaf senescence. A common theme is that HDA9 directly targets and represses the expression of negative regulators, which in turn, promotes leaf senescence (*Figure 7*).

## HDA9 preferentially targets the promoters of actively transcribed genes

HDACs are generally considered as transcriptional co-repressors associated with silent genes. Unexpectedly, our genome-wide occupancy study reveals a predominant enrichment of HDA9 at the actively transcribed genes, and that the binding levels are positively correlated with gene expression (*Figure 3*). Similar genome-wide localization patterns were reported for HDAC1, HDAC2, HDAC3, and HDAC6 in human (*Wang et al., 2009*) and HDA101 in maize (*Yang et al., 2016*). Unlike maize HDA101 that mostly binds transcription start sites (*Yang et al., 2016*), HDA9 is preferentially enriched in the promoters (proximal to transcription start sites) analogous to mammalian HDAC1 and HDAC3 (*Wang et al., 2009*). The positive correlation between HDA9 bound regions and DNase I hypersensitive sites also supports this observation (*Figure 3E*, *Figure 3—figure supplement 1A*). Different from the mammalian and maize studies, we found only a very small fraction of HDA9-enriched genes are silent genes (~5%) (*Figure 3—figure supplement 1C*). Remarkably, none of the 222 HDA9-bound genes that were upregulated in *hda9* are inactive genes (*Figure 3—figure supplement 1C*). The precise mechanism why HDA9 binds to the promoters of actively expressed genes is unclear. One explanation could be that HDA9 may be recruited to the promoters of active genes to prevent promiscuous cryptic transcription. Another possibility is that HDA9 may compete with more active HDAs for binding to the similar genomic regions. It is also unclear why silent genes are not the preferred targets of HDA9. One possibility is that silent genes in *Arabidopsis* tend to have high promoter DNA methylation and thus may be repressed by DNA methylation (unpublished data). Another difference is that our HDA9 affinity purification and MS analysis failed to detect Pol II subunits, which were reported to associate with HDAC in mammals (*Wang et al., 2009*).

We co-purified WRKY53 transcription factor with HDA9. WRKY53 is known to act either as a transcriptional activator or transcriptional repressor in leaf senescence (*Miao et al., 2004*; *Miao and Zentgraf, 2007*; *Miao et al., 2013*). Supporting the relevance of HDA9-WRKY53 interaction, we found that HDA9 binding peaks are significantly enriched for WRKY binding motifs (*Figure 3G*). It is possible that WRKY53 recruits HDA9 to active genes to remove the acetylation marks added by HATs to maintain their proper expression levels during senescence.

## Non-transcriptional functions of HDA9

Analogous to reports on mammalian and maize HDACs (*Wang et al., 2009*; *Yang et al., 2016*), our ChIP-seq and RNA-seq analyses also reveal that the majority of HDA9 bound genes do not alter gene expression in the absence of HDA9. Although the mechanism is unclear, several possibilities could account for this pattern. First, the ultimate gene expression level is determined by the combined actions of multiple redundant repressors or activators, exemplified by yeast Rpd3 and chromatin remodeling enzyme Isw2 (*Fazzio et al., 2001*). It is possible that other HDACs (e.g. HDA6 and HDA19) or transcriptional repressors function redundantly with HDA9 to regulate gene expression, and thus loss of HDA9 itself is insufficient to release the transcriptional repression of its targets. This may also provide an explanation for the low number of mis-regulated genes in *hda6* (*To et al., 2011*; *Blevins et al., 2014*). In support of this notion, loss-of-function *hda6* mutations induce delayed leaf senescence and repress the expression of *SAG12* and *SEN4* (*Wu et al., 2008*). The molecular basis of HDA6 in leaf senescence is unclear. It will be interesting to explore the precise relationship between HDA9 with HDA6 in targeting and regulating gene expression in leaf senescence as well as other biological processes. Second, our GO analysis reveals that HDA9-bound genes that have no significant change in RNA transcripts in *hda9* are enriched for various developmental processes and environmental responses (*Figure 3—figure supplement 1D*). It is possible that the association of HDA9 with the promoters of these developmental genes can rapidly induce the repression of their expression in response to certain internal and/or external signals. Indeed, many photosynthesis-related genes are reported to be upregulated in *hda9* seeds during imbibition (*van Zanten et al., 2014*). Third, HDA9 may act to regulate chromatin structure rather than acting

through transcriptional regulation. We found that HDA9 binds 177 chromatin factors, including SWI/SNF chromatin remodelers, ATP-dependent helicases, methyl DNA binding proteins, Tudor/PWWP/MBT superfamily proteins, and WD-40 proteins (*Figure 3—source data 1C*). Fourth, HDA9 may be recruited to the promoters of active genes to prevent promiscuous cryptic transcription as suggested by mammalian and maize studies (*Wang et al., 2009*; *Yang et al., 2016*). Finally, some of the HDA9 enriched loci may not be its *bona fide* targets. Further experiments will be required to determine the precise function and mechanism of HDA9 in the regulation of its target genes.

## Molecular action of PWR on HDA9 activity in leaf senescence

Our results suggest that HDA9, PWR and the WRKY53 transcription factor form a previously unknown complex that promotes leaf senescence. This complex may be analogous to HDAC3-SMRT/N-CoR repressor complex in animals. Although the function of HDAC3-SMRT/N-CoR in various cellular processes (e.g. development, differentiation, and diseases) has been well studied (*Karagianni and Wong, 2007*; *Shahbazian and Grunstein, 2007*), its role in cellular senescence and aging remains unclear in mammals. Our ChIP assay showed that PWR and HDA9 are enriched at the same loci, and HDA9 binding to these loci requires PWR in vivo (*Figure 6A,D*, *Figure 6—figure supplement 1C*). This is consistent with the role of SMRT/N-CoR in targeting HDAC3 to chromatin (*Guenther et al., 2000*; *Karagianni and Wong, 2007*; *You et al., 2013*). SMRT/N-CoR has an additional function in stimulating HDAC3 deacetylase activity in vitro (*Guenther et al., 2001*; *Watson et al., 2012*). Despite extensive testing, we have been unable to find in vitro conditions that allow the robust HDA9 deacetylase activity. Thus, it remains to be determined whether PWR promotes HDA9 catalytic activity. On the other hand, we discovered a potential role of PWR in HDA9 nuclear localization (*Figure 6E*). Contrast to a previous study showing that HDA9 predominantly localized in the nucleus (*Kang et al., 2015*), we found a significant fraction of HDA9 protein is present in the cytoplasm (*Figure 6E*). Such difference in observation may be due to the fact that mature leaves were examined in our experiment whereas the 10-day-old seedlings were used in the previous study. It will be interesting to examine whether HDA9 localizes differently in the cell during the different developmental stages.

PWR was previously reported to promote the expression of several miRNA genes (*Yumul et al., 2013*). MiRNA is also implicated in leaf senescence (*Kim et al., 2009a*; *Humbeck, 2013*; *Li et al., 2013*; *Huo et al., 2015*). We wonder whether the induction of senescence by PWR also partially depends on miRNA pathways. Although we cannot completely rule out this possibility, several lines of evidence suggest it is less likely. First, the expression of miRNA genes reported in (*Yumul et al., 2013*) is not affected in *hda9* mutants according to our RNA-seq analysis (*Figure 5—source data 1*). Second, *hda9 pwr* double mutant shows no further delay in leaf senescence than any of the *hda9* or *pwr* single mutants, suggesting that PWR and HDA9 act through the same pathway in promoting leaf senescence. Additionally, senescence-related genes regulated by HDA9 and PWR are largely the same group of genes (*Figure 5—source data 3*). Thus, PWR promotes leaf senescence likely through its functional interaction with HDA9.

Taken together, our data provide molecular insights into the function and mechanism of how HDA9-PWR-WRKY53 complex integrates and coordinates multiple signaling pathways to regulate global gene expression during leaf senescence.

## Materials and methods

### Plant materials

*Arabidopsis thaliana* ecotype Columbia-0 (Col-0) was used as wild type (WT) for all experiments. The T-DNA insertion lines of *hda9* (SALK_007123) and *pwr* (SALK_071811C) were obtained from the *Arabidopsis* Biological Resource Center (ABRC). Seeds were sown in soil and kept at 4°C for 2 days before transferring to 24 hr constant light at 22°C.

### Construction of vectors and generation of transgenic plants

The full-length cDNA of *WRKY53* was amplified and cloned into GST tagged protein expression vector pGOOD, modified from pGEX-6P by adding 6XHIS tag at the C terminus. Genomic DNA of PWR and HDA9 with their 1 kb promoters were amplified and cloned into pENTR/D-TOPO. These

constructs were recombined into the pEarleyGate302 binary vectors (*Du et al., 2012*) to create epitope-tagged FLAG or HA fusions and were transformed by agrobacterium-mediated infection into *hda9* mutants or wild type plants. Detailed information for primers can be found in *Supplementary file 1*.

## Dark treatment and chlorophyll content measurement

Rosette leaves were detached from 2-week-old plants and placed on moisturized filter paper in petri dishes. The dishes were kept in dark or constant light at 22°C for 4–5 days. Chlorophyll was extracted from leaves of dark-treated or untreated controls using 80% acetone. Briefly, the leaves were crushed in 1 ml 80% acetone and kept in dark at 4°C overnight. The chlorophyll content was determined as described previously (*Inskeep and Bloom, 1985*) and then normalized to leaf fresh weight.

## RNA extraction and quantitative RT-PCR

For RNA-seq, RNA was extracted from the young leaves (YL) of 10-day-old seedlings and leaves of early senescent plants (ES) grown in constant light at 22°C. Total RNA was extracted with Trizol reagent (ThermoFisher) and treated with DNase I (NEB). One microgram RNA was reverse-transcribed into cDNA with SuperScript III (ThermoFisher) followed by quantitative PCR assay with SYBR Green Master Mix using CFX96 Real-Time System 690 (Bio-Rad, Hercules, CA). Relative transcript level to *ACTIN7* was calculated with the $2^{-\Delta CT}$ method (*Livak and Schmittgen, 2001*). Detailed information for primers can be found in *Supplementary file 1*.

## Western blot

FLAG and HA epitope tags were detected with horseradish peroxidase (HRP) conjugated anti-FLAG (Sigma, A8592) and anti-HA (Roche, 12013819001), respectively. The α-tubulin antibody is from cell signaling (3878). The following histone antibodies were used: H3K9ac (Millipore, 07–352), H3K27ac (Active Motif, 39133), H3ac (Active Motif, 39139), H3 (Abcam, ab1791), H4K8ac (Millipore, 07–328), H4K12ac (Millipore, 07–595), H4K16ac (Millipore, 07–329), H4ac (Active Motif, 39243), H4 (Abcam, ab7311). All western blots were developed using ECL Plus Western Blotting Detection System (GE healthcare, RPN2132).

## Library construction, sequencing, and data analysis

RNA-seq and ChIP-seq libraries were constructed using a TruSeq RNA Library Preparation Kit (Illumina, #RS-122–2002) and the Ovation Ultralow DR Multiplex System (NuGEN, #0330), respectively. Libraries were sequenced on a HiSeq2000 in the UW-Madison Biotechnology Center. Reads were aligned to the *Arabidopsis* reference genome (TAIR10) using Bowtie2 (v2.1.0) with default parameters. Reads that mapped to identical positions in the genome were collapsed into one read. Tophat (2.0.8b) and Cufflink (2.1.1) were used for differential expression analysis (*Trapnell et al., 2012*). The genes showing a p<0.05 were considered as significantly differentially expressed genes. Two biological replicates were performed for RNA-seq. Gene Ontology analysis was performed using agriGO (http://bioinfo.cau.edu.cn/agriGO/). MACS (1.4.2) was used for peak calling with p=1e-03. BEDTools (2.17.0) and custom PERL scripts were used for further analysis. Increased H3K27ac peaks were identified by calling peaks in *hda9* or *pwr* over WT with p=1e-03. In HDA9 binding profiling, $log_2$ value of normalized HDA9 ChIP reads divided by WT reads was calculated and binned in 100 kb intervals. DNase I hypersensitive (DH) sites were from (*Zhang et al., 2012*). All statistical analysis and figures were done using R (3.2.3). The total reads obtained for each sample are listed in *Supplementary file 2*.

## Chromatin immunoprecipitation (ChIP)

Histone ChIP was performed as previously described (*Lu et al., 2015*). A two-gram mixture of mature leaves and early senescent leaves were ground into powder in liquid nitrogen and cross-linked in Nuclei Isolation Buffer I (10 mM Hepes pH 8, 1M Sucrose, 5 mM KCl, 5 mM $MgCl_2$, 5 mM EDTA, 0.6% Triton X-100, 0.4 mM PMSF, and protease inhibitor cocktail tablet [Roche, 14696200]) with 1% formaldehyde for 20 min at room temperature. The homogenate was filtered through two layers of miracloth (Millipore, 475855) and pelleted by centrifuging at 4000 rpm for 25 min at 4°C.

The pellet was washed with Nuclei Isolation Buffer II (0.25 M sucrose, 10 mM Tris-HCl pH 8, 10 mM MgCl$_2$, 1% Triton X-100, 1 mM EDTA, 5 mM β-mercaptoethanol, 0.4 mM PMSF, protease inhibitor cocktail tablet), then resuspended with Nuclear Lysis Buffer (50 mM Tris-HCl pH 8, 10 mM EDTA, 1% SDS, 0.4 mM PMSF, protease inhibitor cocktail tablet) and kept on ice for 10 min. The lysates was diluted 10-fold with ChIP Dilution Buffer (1.1%Triton X-100, 1.2 mM EDTA, 16.7 mM Tris-HCl pH 8, 167 mM NaCl, 0.4 mM PMSF, and protease inhibitor cocktail tablet) and sheared by sonication. After centrifugation at 5000 rpm for 10 min, the supernatant was incubated with 5 µg antibody and 40 µl magnetic protein A/G beads (Life Technologies) overnight with rotation at 4°C. After sequential washes with Low salt Buffer (150 mM NaCl, 0.1% SDS, 1% Triton X-100, 2 mM EDTA, 20 mM Tris-HCl pH 8), High Salt Buffer (500 mM NaCl, 0.1% SDS, 1% Triton X-100, 2 mM EDTA, 20 mM Tris-HCl pH 8), LiCl Buffer (0.25M LiCl, 1% NP-40, 1% sodium deoxycholate, 1 mM EDTA, 10 mM Tris-HCl pH 8) and TE Buffer (10 mM Tris-HCl pH 8, 1 mM EDTA), the DNA-protein complex was eluted with ChIP Elution Buffer (1% SDS, 0.1M NaHCO$_3$) and reverse cross-linked at 65°C for over 6 hr. After proteinase K and RNase treatment, DNA was purified by standard phenol–chloroform method for qPCR. Antibodies used in histone acetylation ChIP were the same as used in western blot.

HDA9-FLAG and PWR-FLAG ChIP were slightly modified from (**Du** *et al., 2012*). Nuclei were isolated from the two grams of a mixture of mature leaves and early senescent leaves and cross-linked in vitro using the same method as described above. After wash with Nuclei Isolation Buffer II, the nuclei was resuspended with IP Binding Buffer (50 mM Tris-HCl pH 8, 150 mM NaCl, 5 mM MgCl$_2$, 5% Glycerol, 0.1% NP-40, 1 mM PMSF, and protease inhibitor cocktail tablet) and sheared by sonication. After centrifugation at 5000 rpm for 10 min, the supernatant was incubated with Anti-FLAG M2 magnetic beads (Sigma, M8823) overnight with rotation at 4°C. After wash with IP Binding Buffer containing 500 mM NaCl, the protein-DNA complex was eluted with ChIP Elution Buffer and reverse cross-linked at 65°C for 6 hr. After proteinase K and RNase treatment, DNA was purified by standard phenol–chloroform method for qPCR analysis or sequencing.

## Affinity purification and mass spectrometry analysis

Affinity purification and mass spectrometry analysis of HDA9-FLAG and PWR-FLAG were performed as previously described (**Du et al., 2012**). Approximately twenty grams of leaves from HDA9-FLAG, PWR-FLAG or WT (negative control) were ground into powder and homogenized in 80 ml IP Binding Buffer (50 mM Tris-HCl pH 8, 150 mM NaCl, 5 mM MgCl$_2$, 5% Glycerol, 0.1% NP-40, 1 mM DTT, 1 mM PMSF, and protease inhibitor cocktail tablet). After centrifugation at 10,000 g for 15 min, the supernatant was incubated with anti-FLAG M2 magnetic beads (Sigma, M8823) with rotation at 4°C for 3 hr. The bead-bound complex was washed 4 times with IP Binding Buffer at 4°C for 5 min each. Bound protein was released by two 10 min incubations with Elution Buffer (50 mM Tris-HCl pH 8, 150 mM NaCl, 5 mM MgCl$_2$, 5% Glycerol, 0.5 mM DTT, 1 mM PMSF, and protease inhibitor cocktail tablet) containing 150 ng/µl 3×FLAG peptide (Sigma, F4799). The eluted protein complexes were precipitated with trichloroacetic acid, further digested with Trypsin, and analyzed on an Orbitrap mass spectrometer (LTQ Velos, ThermoFisher Scientific).

HPLC separation employed a 100 x 365 µm fused silica capillary micro-column packed with 20 cm of 1.7µm-diameter, 130 Angstrom pore size, C18 beads (Waters BEH), with an emitter tip pulled to approximately 1 µm using a laser puller (Sutter instruments). Peptides were loaded on-column at a flow-rate of 400 nL/min for 30 min and then eluted over 120 min at a flow-rate of 300 nl/min with a gradient of 2% to 30% acetonitrile, in 0.1% formic acid. Full-mass profile scans were performed in the FT orbitrap between 300–1500 m/z at a resolution of 60,000, followed by ten MS/MS HCD scans of the ten highest intensity parent ions at 42% relative collision energy and 7500 resolution, with a mass range starting at 100 m/z. Dynamic exclusion was enabled with a repeat count of two over the duration of 30 s and an exclusion window of 120 s.

For data analysis, the acquired precursor MS and MS/MS spectra were searched against a Mus musculus protein database (Uniprot reviewed canonical database, containing 16,639 sequences) using SEQUEST, within the Proteome Discoverer 1.3.0.339 software package (ThermoFisher Scientific). Masses for both precursor and fragment ions were treated as mono-isotopic. Oxidized methionine (+15.995 Da) and the gly-gly footprint on lysine (+114.043 Da) were allowed as dynamic modifications and carbamidomethylated cysteine (+57.021 Da) was searched as a static modification. The database search permitted for up to two missed trypsin cleavages and ion masses were

matched with a mass tolerance of 10 ppm for precursor masses and 0.1 Da for HCD fragments. The output from the SEQUEST search algorithm was validated using the Percolator algorithm. The data were filtered using a 1% false discovery rate ( *Rohrbough et al., 2006*), based on q-Values, with a minimum of two peptide matches required for confident protein identification.

## Co-immunoprecipitation analysis

Co-immunoprecipitation was performed in 1.5 g F1 *Arabidopsis* plants co-expressing HDA9-HA and PWR-FLAG. Total extracts were incubated with 25 µl FLAG magnetic beads for PWR-FLAG purification and the HDA9-HA was detected by using anti-HA-Peroxidase High Affinity 3F10 antibody (Roche, 13800200).

## GST protein purification and pull down assay

GST tag only and GST-WRKY53 proteins were induced with 200 µM IPTG (Isopropyl β-D-1-thiogalactopyranoside) for three hours at 37°C. HDA9-FLAG protein was purified from HDA9-FLAG transgenic plants using IP methods described above. Purified HDA9-FLAG and GST tagged protein were incubated in the pull down buffer (20 mM Tris-HCl 8.0, 200 mM NaCl, 1 mM EDTA, 0.5% Nonidet P-40) for one hour then incubated with Glutathione Sepharose 4B beads (GE Healthcare, 17075601) for one hour at 4°C. After washed with pull down buffer for three times with 5 min per wash, protein-bead complex was boiled in SDS loading buffer, subjected to SDS-PAGE gel, and detected with anti-GST and anti-FLAG antibodies.

## Nuclear-cytoplasmic fractionation

Mature leaves of HDA9-FLAG/*hda9* and HDA9-FLAG/*hda9 pwr* plants were used. Nuclear-cytoplasmic fractionation of HDA9-FLAG was performed as previously described (*Wang et al., 2011a*).

## Accession codes

RNA-seq and ChIP-seq data were deposited into GEO with the accession number GSE80056.

## Acknowledgements

We thank the UW-Madison Biotech Center for high-throughput sequencing. We are grateful to Richard Amasino, John Denu, Rupa Sridharan, and members of X Zhong laboratory for discussions and critical comments on this manuscript. Work in XZ's laboratory was supported by startup funds from UW-Madison, an NSF CAREER award (MCB-1552455), Alexander von Humboldt Foundation (Alfred Toepfer Faculty Fellow), and USDA and National Institute of Food and Agriculture grant (Hatch 1002874). Work in LMS's laboratory was supported by NIH grants (1RO1GM114292).

## Additional information

### Funding

| Funder | Grant reference number | Author |
| --- | --- | --- |
| National Institutes of Health | 1RO1 GM114292 | Lloyd M Smith |
| University of Wisconsin-Madison | Startup grant | Xuehua Zhong |
| National Science Foundation | CAREER award, MCB-1552455 | Xuehua Zhong |
| USDA and National Institute of Food and Agriculture | Hatch 1002874 | Xuehua Zhong |
| Alexander von Humboldt-Stiftung | Alfred Toepfer Faculty Fellow | Xuehua Zhong |

The funders had no role in study design, data collection and interpretation, or the decision to submit the work for publication.

## Author contributions

XC, Acquisition of data, Analysis and interpretation of data, Drafting or revising the article; LL, Analysis and interpretation of data; KSM, MS, SQ, AL, LMS, Acquisition of data; XZ, Conception and design, Analysis and interpretation of data, Drafting and revising the article

## Author ORCIDs

Xuehua Zhong, http://orcid.org/0000-0002-2350-0046

## Additional files

### Supplementary files

• Supplementary file 1. Primers used in this study.

• Supplementary file 2. Reads of RNA-Seq and ChIP-Seq.

### Major datasets

The following dataset was generated:

| Author(s) | Year | Dataset title | Dataset URL | Database, license, and accessibility information |
|---|---|---|---|---|
| Xuehua Zhong, Xiangsong Chen, Li Lu | 2016 | Functional analysis of HDA9 in Arabidopsis | https://www.ncbi.nlm.nih.gov/geo/query/acc.cgi?acc=GSE80056 | Publicly available at the NCBI Gene Expression Omnibus (accession no: GSE80056) |

The following previously published dataset was used:

| Author(s) | Year | Dataset title | Dataset URL | Database, license, and accessibility information |
|---|---|---|---|---|
| Wenli Zhang, Tao Zhang, Yufeng Wu, Jiming Jiang | 2012 | Mapping regulatory elements using signatures of open chromatin in Arabidopsis thaliana | https://www.ncbi.nlm.nih.gov/geo/query/acc.cgi?acc=GSE34318 | Publicly available at the NCBI Gene Expression Omnibus (accession no: GSE34318) |

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
