## [Decision Letter]

[Editors’ note: this article was originally rejected after discussions between the reviewers, but the authors were invited to resubmit after an appeal against the decision.]

Thank you for submitting your work entitled "POWERDRESS recruits HISTONE DEACETYLASE 9 to chromatin to promote ageing in *Arabidopsis*" for consideration by *eLife*. Your article has been evaluated by Jessica Tyler (Senior Editor) and two reviewers, one of whom is a member of our Board of Reviewing Editors. The reviewers have opted to remain anonymous.

Our decision has been reached after consultation between the reviewers. Based on these discussions and the individual reviews below, we regret to inform you that your work will not be considered further for publication in *eLife*.

As you will see, both of the reviewers expressed concerns about how the data were interpreted in the manuscript, as well as the lack of supporting evidence for some conclusions. The reviewer 1 also pointed out a serious disconnect between the genome-wide data sets and the specific analysis for genes related to leaf senescence. These issues significantly weakened the manuscript.

Reviewer #1:

The manuscript by Chen et al. used combined biochemical, genomic, genetic and molecular approaches to implicate a SANT domain protein and a histone deacetylase in joint promotion of senescence. In addition, a WRKY transcription factor is implicated in tethering the co-repressor complex to target loci. Regulation of chromatin modifying enzyme activity is a topic of broad general interest of relevance for development and response to the environment alike. My enthusiasm for this study was dampened however by the lack of logical overlap between the individual findings as outlined below.

1) My biggest concern is that none of the genome-wide datasets produced are linked to the detailed molecular analyses. The authors identify 277 genes that are upregulated in both *hda9* and *pwr* mutants and then go on to study in detail the impact of HDA9 and PWR on four genes with a role in leaf senescence (WRKY57, APG9, CAT1, NPX1 (Figure 5, Figure 6)). However, none of these four genes are actually in the 277 gene list. I found Figure 5 in particular very misleading in this regard. This begs the question what the purpose of the genomic datasets presented is – if in the end known candidate genes are studied. In addition, when I checked which of the 277 genes upregulated in both *hda9* and *pwr* are direct HDA9 targets, I found that only roughly one third are (91 genes). Given that HDA9 binds 8000/24000 loci, the 277 gene list is thus not enriched in HDA9-bound genes. Likewise, the genes upregulated in *hda9* mutants are not enriched in HDA9 bound loci.

2) This leads me to the second concern. Surprisingly the authors find that –genome-wide – HDA9 binds to actively expressed genes, most of which are not differentially expressed in *hda9* mutants. While the latter can be attributed to redundantly acting HDACs, this nevertheless raises the question why HDA9 is present at active loci. Of note, the authors also report a global reduction in two active marks H3K9ac and H3K27ac in *hda9* mutants on Westerns. One explanation could be that HDA9 resides at genomic loci in an inactive, latent form that needs to be activated or unblocked to have enzymatic activity. Another possibility is that HDA9 binding may prevent binding of other, more active HDAs, to these loci. In either case, one would expect high levels of H3K9ac or H3K27ac at the HDA9 bound highly expressed loci. This should be tested- either genome-wide or at select HDA9 targets in this category. If indeed H3K9ac and H3K27ac levels are high at HDA9-bound loci, HDA9 binding is not directly linked to altered transcriptional output. A complementary approach, monitoring changes in histone acetylation levels genome-wide in *hda9* and *pwr* mutants relative to the WT, might also help to understand the role of HDA9 in gene regulation.

3) The title claims that PWR recruits HDA9 to chromatin. This conclusion needs further supporting data – for example PWR binding to the same region as HDA9 on HDA9 targets on the basis of ChIP -, given that HDA9-FLA levels are reduced in the nucleus of *pwr* plants, It seems the authors have tagged PWR plant lines in hand.

Reviewer #2:

In the manuscript titled "POWERDRESS recruits HISTONE DEACETYLASE 9 to chromatin to promote ageing in *Arabidopsis*", the authors identified physical association between histone deacetylase HDA9 and PWR and characterized their function by new genome-wide analysis including ChIP-seq and RNA-seq. These results suggest that PWR recruits HDA9 to its genomic targets and functions to promote leaf senescence associated with age and darkness through transcription factor WRKY53.

This manuscript reports a novel function for HDA9 and PWR in regulating leaf senescence associated with age and darkness in *Arabidopsis*, and should be of interest for the readers of *eLife*. Overall the experiments were well designed and executed. The data presented are clear and support their conclusions. However, there are a few issues that should be further clarified before the manuscript is accepted for publication.

1) After the HDA9 ChIP-seq and *hda9*∆ RNA-seq analysis, the authors found that 95% of genes bound by HDA9 did not change gene expression when HDA9 is deleted. The authors interpreted it as either HDA9 is not a primary regulator of transcription for the majority of its targets or it is insufficient to regulate transcription. I think that it is most likely that gene expressions are regulated and repressed by multiple redundant or parallel repressors and eliminating one of them is usually not enough to derepress and activate transcription. There are such examples clearly described in yeast between HDAC Rpd3 and chromatin remodeling enzyme Isw2 (Fazzio et al., MCB, 2001).

2) The authors identified WRKY binding motif in HDA9 target sites. However, it is not clear whether the "target sites" simply means HDA9 ChIP-seq peak or it means those genes that both have HDA9 ChIP-seq peaks and altered gene expression in *hda9*. In either case, the exact numbers of genes containing the WRKY binding motif should be given with a p-value.

3) The authors compared the up-regulated and down-regulated genes in *hda9* and *pwr* mutants and found substantial overlaps. Fisher's exact test p-values should be given for this analysis to show whether the overlap was significantly higher than by chance.

4) The authors showed four genes in Figure 6 as "direct targets" of HDA9. What exactly does it mean? Does it mean both HDA9 ChIP-seq peak and increased expression in *hda9* by RNA-seq?

5) Since the model is PWR recruiting HDA9 to target genes, one would predict that genes with both HDA9 ChIP-seq peaks and up-regulated by *hda9* mutation should show higher chance of PWR binding. This would provide further support for the model if proven true.

[Editors’ note: what now follows is the decision letter after the authors submitted for further consideration.]

Thank you for resubmitting your work entitled "POWERDRESS recruits HISTONE DEACETYLASE 9 to chromatin to promote aging in *Arabidopsis*" for further consideration at *eLife*. Your revised article has been favorably evaluated by Jessica Tyler (Senior Editor), a Reviewing Editor, and 1 reviewer.

The manuscript has been improved but there are some remaining issues that need to be addressed before acceptance, as outlined in the review below. Point 1 cannot really be addressed, however, given your response to the previous comments and the clarification of candidate gene selection criteria now added to the manuscript, it will be fine to leave it as is. Please pay attention to address the points 2 and 3 in your revision.

Reviewer #2:

The revised manuscript is much improved by the additional data included.

I have a couple of comments and clarifications for the points I had raised before.

1) Overlap of genome-wide and gene-specific analyses

The new logic presented in Figure 5 and Figure 5—figure supplement 1 clarifies things. The fact remains that the authors elected to focus on three genes, of which, on the basis of the genome-wide analyses, two are direct HDA9 targets (bound and upregulated in *hda9*) and one of the two is also upregulated in *pwr* mutants (NPX1).

2) Role of HDA9 at active loci. Adding the H3K27ac ChIPseq strengthens the manuscript and, in particular, helps solidify the link between PWR and HDA9. In addition, HDA9 bound genes have higher H3K27ac than nonbound genes, supporting the idea that HDA9 resides at active loci. In *hda9* and *pwr* mutants there is perhaps a stronger increase in H3K27ac at HDA9 bound loci than at nonbound loci. Is that true? This could be tested statistically. One remaining question is how do the changes in H3K27ac in the mutant compared to the wild type link to the changes in gene expression in the same genetic backgrounds? The authors should include a Venn diagram for the overlap between genes bound by HDA9, upregulated in *hda9* and genes with significantly increased H3K27ac in *hda9*. This is very easily done with the available data.

Also, I apologize for a typo in my comment: should have read " a global increase in two active marks".

3) PWR recruits HDA9. The additional PWR ChIP data is nice. However, the authors need to rephrase their conclusions. Nuclear HDA9 levels are reduced in *pwr* (HDA9-FLAG *hda9 pwr*
Figure 6), the genetic background used for ChIP-qPCR (Figure 6, Figure 6—figure supplement 1). Because the nuclear HDA9 levels are lower it is not clear whether the reduced HDA9 chromatin occupancy in pwr is due to failure to import HDA9 into the nucleus, failure to recruit HDA9 to genomic loci or both.

---

## [Author Response]

[Editors’ note: the author responses to the first round of peer review follow.]

[…]

As you will see, both of the reviewers expressed concerns about how the data were interpreted in the manuscript, as well as the lack of supporting evidence for some conclusions. The reviewer 1 also pointed out a serious disconnect between the genome-wide data sets and the specific analysis for genes related to leaf senescence. These issues significantly weakened the manuscript.

*Reviewer #1:*

The manuscript by Chen et al. used combined biochemical, genomic, genetic and molecular approaches to implicate a SANT domain protein and a histone deacetylase in joint promotion of senescence. In addition, a WRKY transcription factor is implicated in tethering the co-repressor complex to target loci. Regulation of chromatin modifying enzyme activity is a topic of broad general interest of relevance for development and response to the environment alike. My enthusiasm for this study was dampened however by the lack of logical overlap between the individual findings as outlined below.

1) My biggest concern is that none of the genome-wide datasets produced are linked to the detailed molecular analyses. The authors identify 277 genes that are upregulated in both hda9 and pwr mutants and then go on to study in detail the impact of HDA9 and PWR on four genes with a role in leaf senescence (WRKY57, APG9, CAT1, NPX1 (Figure 5, Figure 6)). However, none of these four genes are actually in the 277 gene list. I found Figure 5 in particular very misleading in this regard. This begs the question what the purpose of the genomic datasets presented is – if in the end known candidate genes are studied. In addition, when I checked which of the 277 genes upregulated in both hda9 and pwr are direct HDA9 targets, I found that only roughly one third are (91 genes). Given that HDA9 binds 8000/24000 loci, the 277 gene list is thus not enriched in HDA9-bound genes. Likewise, the genes upregulated in hda9 mutants are not enriched in HDA9 bound loci.

We appreciate these comments and apologize for the confusion. Of the four selected genes, *NPX1* (AT5G63320) is indeed among the 277 genes that are upregulated in both *hda9* and *pwr* mutants ([Supplementary-material SD5-data]). We chose specific genes for further studies when they fulfilled all three of the following criteria: a) they are bound by HDA9 by ChIP-seq ([Supplementary-material SD2-data]); b) they are significantly upregulated in *hda9* by RNA-seq ([Supplementary-material SD3-data]); and c) their roles in leaf senescence. Thus, our molecular analysis would be on HDA9 directly bound targets that had a transcriptional response in the senescence phenotype. Using these three criteria, we narrowed down to 11 genes and focused on four genes (*WRKY57*, *APG9, CAT1, NPX1*) for further investigation. We have now presented this selection strategy as a new Figure 5—figure supplement 1 for clarity. Three genes (*WRKY57*, *APG9, NPX1*) are also upregulated in the *pwr* mutant. While *NPX1* is significantly upregulated (Student’s t test, p<0.05), *WRKY57* and *APG9* are not defined to be significant based on our significance criteria. Thus, we performed RT-qPCR to further confirm their upregulation in *pwr* with additional biological replicates(Figure 5).

We believe that HDA9, like the other HDACs, is involved in many pathways. Leaf senescence is one of its multiple roles. The 277 genes upregulated in both *hda9* (total of 782, 35%) and *pwr* (total of 887, 31%) are indeed significant (Fisher’s exact test, p<2.2e-16). Our GO analysis of differentially expressed genes in both *hda9* and *pwr* shows their significant enrichment in senescence related pathways (Figure 5—figure supplement 1), consistent with our phenotypic and molecular analyses of HDA9 and PWR function in leaf senescence (Figure 4 and Figure 5). Thus, we feel that our genomic data is useful to guide the functional studies.

The fact that 91 out of 277 genes (~33%) upregulated in both *hda9* and *pwr* are direct HDA9 targets is intriguing as HDA9 binds and regulates many transcription and chromatin factors, thus HDA9 may be at the top of a regulatory network ([Supplementary-material SD2-data]). Two-third of these 277 upregulated genes may be indirectly regulated by HDA9 via these factors.

We agree with this review that upregulated genes in *hda9* are not significantly enriched in HDA9-bound genes. As discussed in this manuscript, 95% of ~8,000 HDA9 binding genes do not exhibit a significant change in gene expression. We have further clarified it in our Discussion section, as illustrated below.

“Analogous to reports on mammalian and maize HDACs (Wang et al. 2009; Yang et al. 2016), our ChIP-seq and RNA-seq analyses also reveals that the majority of HDA9 bound genes do not alter gene expression when HDA9 is deleted. […]It is possible that other HDACs (e.g. HDA6 and HDA19) or transcriptional repressors function redundantly with HDA9 to regulate gene expression, and thus loss of HDA9 itself is insufficient to release the transcriptional repression of its targets.”.

2) This leads me to the second concern. Surprisingly the authors find that –genome-wide – HDA9 binds to actively expressed genes, most of which are not differentially expressed in hda9 mutants. While the latter can be attributed to redundantly acting HDACs, this nevertheless raises the question why HDA9 is present at active loci. Of note, the authors also report a global reduction in two active marks H3K9ac and H3K27ac in hda9 mutants on Westerns. One explanation could be that HDA9 resides at genomic loci in an inactive, latent form that needs to be activated or unblocked to have enzymatic activity. Another possibility is that HDA9 binding may prevent binding of other, more active HDAs, to these loci. In either case, one would expect high levels of H3K9ac or H3K27ac at the HDA9 bound highly expressed loci. This should be tested- either genome-wide or at select HDA9 targets in this category. If indeed H3K9ac and H3K27ac levels are high at HDA9-bound loci, HDA9 binding is not directly linked to altered transcriptional output. A complementary approach, monitoring changes in histone acetylation levels genome-wide in hda9 and pwr mutants relative to the WT, might also help to understand the role of HDA9 in gene regulation.

As suggested, we generated H3K27ac ChIP-seq in *hda9* and *pwr* mutants and showed global increase of H3K27ac in both mutants compared to the WT. These new data have been included in the Results section, as illustrated below.

“Consistent with immunoblotting, we identified 11,372 and 7,687 H3K27ac increased peaks in *hda9* and *pwr*, respectively (Figure 2). […] Together, these results suggest that PWR and HDA9 mediate deacetylation of H3K27ac at similar genomic regions.”

We do not know exactly why HDA9 binds to the promoters of actively expressed genes. We appreciate reviewer’s suggestion and have included these possible mechanisms in the Discussion section, as illustrated below.

“The precise mechanism why HDA9 binds to the promoters of actively expressed genes is unclear. One explanation could be that HDA9 may be recruited to the promoters of active genes to prevent promiscuous cryptic transcription. Another possibility is that HDA9 may compete with more active HDAs for binding to the similar genomic loci.”

3) The title claims that PWR recruits HDA9 to chromatin. This conclusion needs further supporting data – for example PWR binding to the same region as HDA9 on HDA9 targets on the basis of ChIP -, given that HDA9-FLA levels are reduced in the nucleus of pwr plants. It seems the authors have tagged PWR plant lines in hand.

As suggested, we performed ChIP-qPCR in PWR-FLAG plants. Additionally, we compared the H3K27ac levels of HDA9 bound genes over non-HDA9 bound genes andfound that *pwr* induced H3K27 hyperacetylation is correlated with HDA9 binding. These new datahave been included in the Results section, as illustrated below.

“We found that 9 of the 11 randomly chosen HDA9 bound loci showed significant enrichment of PWR (Figure 6—figure supplement 1). Furthermore, PWR specifically binds to the same genomic regions within *APG9, WRKY57*, and *NPX1* where HDA9 is enriched (Figure 6). […..]To further examine whether *pwr* induced H3K27 hyperacetylation is correlated with HDA9 binding, we compared the H3K27ac levels of HDA9 bound genes over non-HDA9 bound genes, and found that HDA9 bound genes showed a significantly higher increase of H3K27ac relative to non-HDA9 binding genes in *pwr* (p<2.2e-16) (Figure 6). Together,these results suggest that PWR binds to the same genomic regions as HDA9 on HDA9 targets.”

Reviewer #2:

[…]

This manuscript reports a novel function for HDA9 and PWR in regulating leaf senescence associated with age and darkness in Arabidopsis, and should be of interest for the readers of eLife. Overall the experiments were well designed and executed. The data presented are clear and support their conclusions. However, there are a few issues that should be further clarified before the manuscript is accepted for publication.

1) After the HDA9 ChIP-seq and hda9∆ RNA-seq analysis, the authors found that 95% of genes bound by HDA9 did not change gene expression when HDA9 is deleted. The authors interpreted it as either HDA9 is not a primary regulator of transcription for the majority of its targets or it is insufficient to regulate transcription. I think that it is most likely that gene expressions are regulated and repressed by multiple redundant or parallel repressors and eliminating one of them is usually not enough to derepress and activate transcription. There are such examples clearly described in yeast between HDAC Rpd3 and chromatin remodeling enzyme Isw2 (Fazzio et al., MCB, 2001).

We appreciate reviewer’s suggestion and have included this possible mechanism in the Discussion section, as illustrated below.

“our ChIP-seq and RNA-seq analyses also reveal that the majority of HDA9 bound genes do not alter gene expression when HDA9 is deleted. Although the mechanism is unclear, several possibilities could account for this pattern. […] It is possible that other HDACs (e.g. HDA6 and HDA19) or transcriptional repressors function redundantly with HDA9 to regulate gene expression, and thus loss of HDA9 itself is insufficient to release the transcriptional repression of its targets.”

2) The authors identified WRKY binding motif in HDA9 target sites. However, it is not clear whether the "target sites" simply means HDA9 ChIP-seq peak or it means those genes that both have HDA9 ChIP-seq peaks and altered gene expression in hda9. In either case, the exact numbers of genes containing the WRKY binding motif should be given with a p-value.

We apologize for the confusion. We have clarified it in the text that the DNA-binding motifs were identified based on the HDA9 binding sequences. We have also added the p-value in the text as following: “1,328 HDA9 binding peaks (14%, p=1.9e-8) showed the significant enrichment of W-box motif (TTGAC/T), recognized by WRKY family transcription factors (Figure 3), consistent with our observation that WRKY53 is co-purified with HDA9 (Figure 1).”

3) The authors compared the up-regulated and down-regulated genes in hda9 and pwr mutants and found substantial overlaps. Fisher's exact test p-values should be given for this analysis to show whether the overlap was significantly higher than by chance.

As suggested, we have added Fisher’s exact test p-value in Figure 2 (p<2.2e-16) and Figure 5 (p<2.2e-16).

4) The authors showed four genes in Figure 6 as "direct targets" of HDA9. What exactly does it mean? Does it mean both HDA9 ChIP-seq peak and increased expression in hda9 by RNA-seq?

Direct targets represent the genes that are bound by HDA9 by ChIP-seq and are upregulated in *hda9* by RNA-seq. We have clarified it in the revised manuscript.

5) Since the model is PWR recruiting HDA9 to target genes, one would predict that genes with both HDA9 ChIP-seq peaks and up-regulated by hda9 mutation should show higher chance of PWR binding. This would provide further support for the model if proven true.

Thanks for the suggestions. We have added new data in Figure 6 by performing PWR-FLAG ChIP-qPCR in different genomic regions of HDA9 targets as well as testing PWR enrichment in 11 randomly chosen HDA9 bound loci (Figure 6—figure supplement 1). Additionally, we performed H3K27ac in *pwr* and compared the H3K27ac levels of HDA9 bound genes over non-HDA9 bound genes andfound that *pwr* induced H3K27 hyperacetylation is correlated with genome-wide HDA9 binding (Figure 6). These new datahave been included in the Results section, as illustrated below.

“We found that 9 of the 11 randomly chosen HDA9 bound loci showed significant enrichment of PWR (Figure 6—figure supplement 1). […]Together,these results suggest that PWR binds to the same genomic regions as HDA9 on HDA9 targets.”

[Editors’ note: the author responses to the re-review follow.]

[…]

The manuscript has been improved but there are some remaining issues that need to be addressed before acceptance, as outlined in the review below. Point 1 cannot really be addressed, however, given your response to the previous comments and the clarification of candidate gene selection criteria now added to the manuscript, it will be fine to leave it as is. Please pay attention to address the points 2 and 3 in your revision.

*Reviewer #2:*

The revised manuscript is much improved by the additional data included.

I have a couple of comments and clarifications for the points I had raised before.

*1) Overlap of genome-wide and gene-specific analyses*

The new logic presented in Figure 5 and Figure 5—figure supplement 1 clarifies things. The fact remains that the authors elected to focus on three genes, of which, on the basis of the genome-wide analyses, two are direct HDA9 targets (bound and upregulated in hda9) and one of the two is also upregulated in pwr mutants (NPX1).

Please see our previous responses.

2) Role of HDA9 at active loci. Adding the H3K27ac ChIPseq strengthens the manuscript and, in particular, helps solidify the link between PWR and HDA9. In addition, HDA9 bound genes have higher H3K27ac than nonbound genes, supporting the idea that HDA9 resides at active loci. In hda9 and pwr mutants there is perhaps a stronger increase in H3K27ac at HDA9 bound loci than at nonbound loci. Is that true? This could be tested statistically. One remaining question is how do the changes in H3K27ac in the mutant compared to the wild type link to the changes in gene expression in the same genetic backgrounds? The authors should include a Venn diagram for the overlap between genes bound by HDA9, upregulated in hda9 and genes with significantly increased H3K27ac in hda9. This is very easily done with the available data.

Also, I apologize for a typo in my comment: should have read " a global increase in two active marks".

Thanks for the suggestions. Indeed, we found that HDA9 bound genes showed a significantly stronger increase of H3K27ac relative to non-HDA9 bound genes in both *hda9* and *pwr* (p<2.2e-16) as presented in Figure 6.

As suggested, we performed the overlap analysis between HDA9 ChIP-seq, *hda9* RNA-seq, and *hda9* H3K27ac ChIP-seq datasets. We noted that 151 out of 222 genes (~68%) that bound by HDA9 and upregulated in *hda9* showed significantly increased H3K27ac in *hda9* (p=1.2e-10). This new datahas been included in Figure 5—figure supplement 1.

3) PWR recruits HDA9. The additional PWR ChIP data is nice. However, the authors need to rephrase their conclusions. Nuclear HDA9 levels are reduced in pwr (HDA9-FLAG hda9 pwr Figure 6), the genetic background used for ChIP-qPCR (Figure 6, Figure 6—figure supplement 1). Because the nuclear HDA9 levels are lower it is not clear whether the reduced HDA9 chromatin occupancy in pwr is due to failure to import HDA9 into the nucleus, failure to recruit HDA9 to genomic loci or both.

We appreciate this comment and have changed the title to “POWERDRESS interacts with HISTONE DEACETYLASE 9 to promote aging in *Arabidopsis*”. We have also rephrased the conclusion as illustrated below.

“HDA9 nuclear accumulation and chromatin association are dependent on PWR”.